# Exploring the association between housing insecurity and mental health among renters: A systematic review of quantitative primary and secondary studies

Mira Talmatzky‡*, Laura Nohr‡, Christine Knaevelsrud, Helen Niemeyer

Division of Clinical Psychological Intervention, Department of Education and Psychology, Freie Universität Berlin, Berlin, Germany

‡ These authors share first authorship on this work.
* mira.talmatzky@fu-berlin.de

## Abstract

Adverse social and economic conditions negatively impact mental health and well-being. To the best of the authors' knowledge, the present systematic review is the first to investigate the association between housing insecurity and mental health outcomes among renters, with a focus on housing affordability and instability. We followed the Preferred Reporting Items for Systematic Reviews and Meta-Analyses (PRISMA) guidelines. A comprehensive search was conducted in December 2022 across four databases (MEDLINE, PsycINFO, Web of Science, and ASSIA). Quantitative studies from OECD (Organisation for Economic Co-operation and Development) member countries were eligible for inclusion if they investigated housing insecurity by examining at least one independent variable related to housing affordability and/or instability, and included at least one mental health-related outcome among adult renters. Studies needed to specifically distinguish effects for renters, not simply adjust for tenure or include an interaction. Studies were excluded if they analyzed other forms of housing insecurity, examining residential satisfaction and general health outcomes, or populations other than adult renters (< 15 years of age). The methodological quality of the included studies was rated with the JBI Critical Appraisal Tools, and the certainty of evidence was rated using the Grading of Recommendations Assessment, Development, and Evaluation (GRADE) framework. Due to heterogeneity of the identified studies, we performed a narrative synthesis. Twenty-two studies with sample sizes ranging from $n = 89$ to $n = 179,037$ met the inclusion criteria (resulting in a total sample size of at least $N = 336,775$), of which 14 applied a longitudinal design, five a cross-sectional design, and three a quasi-experimental design. Based on the JBI ratings, the overall methodological quality of the included studies was good. The overall ratings of certainty of evidence, based on the GRADE ratings, were between low and very low – mainly due to the

**Data availability statement:** All relevant data are within the paper and its Supporting Information files.

**Funding:** The author(s) received no specific funding for this work.

**Competing interests:** The authors have declared that no competing interests exist.

non-controlled study designs of included studies. Among the nine studies examining housing affordability, six reported significant associations between unaffordable rent and poor mental health in low-income renters. Regarding housing instability, 12 out of 14 studies reported significant associations between unstable housing and renters' mental health issues. Measures of housing insecurity varied, with rent-to-income ratio and forced moves being most commonly employed. Mental health outcomes focused primarily on overall mental health, well-being, and depressive symptoms, while few studies explored other mental health outcomes. Despite methodological limitations due to the non-controlled studies included in the review, the findings suggest overall that experiencing unaffordable or unstable housing has a negative impact on renters' overall mental health and depressive symptoms. Housing insecurity poses a significant challenge for renters in OECD countries, highlighting the need for policymakers to implement supportive housing policies and tenure protection measures in order to improve renters' housing security and ultimately public health. Nevertheless, more research with robust study designs is needed to draw further conclusions. The systematic review has been conducted without external funding. It has not been pre-registered and no study protocol has been published.

## Introduction

### Background

Social conditions and contextual factors significantly influence the risk, development, and persistence of mental disorders and psychological distress. According to the World Health Organization (WHO), social determinants, encompassing the conditions in which individuals are born, grow, work, live, and age, contribute to severe health inequalities within contemporary societies [1]. Housing is widely recognized as a significant social determinant influencing mental health and contributing to public health challenges [2–8]. Housing insecurity is a global phenomenon, which exposes individuals to various forms of challenges, including unaffordable housing, poor housing conditions, a lack of stable tenure, and homelessness [9–12]. Renters, and particularly low-income renters, are especially vulnerable to these housing disadvantages [9,13,14]. Indeed, insecure housing not only compromises the basic need for shelter but also poses significant risks to mental health.

Multiple systematic reviews have explored the association between housing insecurity and mental health outcomes, providing valuable insights within this field of research. Nevertheless, all of these reviews either did not differentiate by tenure status or focused specifically on tenures other than renting, such as home ownership or homelessness. For instance, research has investigated a range of housing disadvantages, such as eviction and foreclosure [15,16], financial distress of homeowners [17], and homelessness [18,19], and their link to mental health. In a systematic review of longitudinal studies, Singh et al. [20] found that prior exposure to housing disadvantages can have effects on mental health later in life. Moreover, the impact of

living environment and neighborhood conditions, such as poor housing quality and air pollution, on psychological distress has been extensively studied in multiple systematic reviews [21–23]. Additionally, systematic reviews have explored the relationship between housing and mental health in specific populations, such as students [24], children [25,26], and individuals with severe mental illnesses [27]. However, to the best of our knowledge, no review to date has specifically examined the issue of housing insecurity among adult renters and its impact on mental health.

## Linking housing insecurity to renters' mental health

Housing scholars have long emphasized that housing is more than a mere physical shelter. Rather, it constitutes a social and emotional environment that significantly impacts our psychological well-being [28–31]. The acute and chronic stress resulting from exposure to social disadvantages directly and indirectly impact the development and course of mental disorders [5]. In a systematic review of social determinants influencing mental disorders, Lund et al. [4] identified depression, anxiety, substance abuse, suicide, psychosis, and dementia as potential outcomes associated with economic disadvantages, including housing issues. Traditionally, the literature has emphasized the (health) benefits of homeownership, while considering rental tenure as inherently negative for mental well-being due to its lack of stability and ontological security [32,33]. Recently, however, a growing body of research has explicitly examined the composition of cohorts living in rental housing as well as the specific rental conditions that impact mental health [34–37]. A factor driving this shift was the sudden rise in the number of renters, particularly within the private rental sector, in high-income countries traditionally dominated by homeownership, such as Australia [38], the United Kingdom [UK; 39], and the United States of America [U.S.; 40].

Housing insecurity is profoundly influenced by various macrostructural factors, including the welfare state, housing system, public housing policies, and market dynamics [41]. In their paper, Marí-Dell'Olmo et al. [41] provide a conceptual framework on housing systems, housing conditions, and health equity, illustrating the interaction among these structural dimensions and their impact on access to adequate housing and subsequent physical and mental health outcomes. Despite the diverse housing systems, regulatory environments, and socio-economic factors among countries, many advanced economies share a common trajectory of severe housing insecurity challenges [42]. This review focuses on OECD countries for several reasons: They typically exhibit similar economic structures characterized by market-based economies with varying levels of government intervention, and they often adopt comparable housing policies and regulations, albeit with variations in their scope and implementation, such as social housing programs and tenant protections. Furthermore, in many OECD countries, heavy reliance on the market for access to housing has led to rising prices and reduced accessibility of housing, exacerbating the challenges for renters in securing affordable, stable, and suitable housing [2,10,14,39,43]. These challenges have been further reinforced by recent societal upheavals such as the COVID-19 pandemic, related social and economic lockdowns, and the energy crisis [44–46]. Moreover, public expenditure on housing remains low across many OECD countries, resulting in a heavy reliance on the often minimally regulated private rental market for access to affordable housing [47]. Although some governments have implemented a range of housing policies and programs to address the issue of housing security, including social/public housing, housing subsidies, or eviction policies, these initiatives frequently fail to reach those who truly need them, leaving many eligible renters without the necessary support [9,48]. In summary, renters have emerged as a vulnerable group that is disproportionately affected by housing insecurity and economic crises. Against this background, the present systematic review aims to explore the association between housing insecurity and mental health outcomes among adult renters within OECD countries.

## Conceptualization of housing insecurity among renters

Housing insecurity is a multidimensional concept. Although definitions vary across scholars, commonly applied dimensions include housing affordability, quality, (in-)stability, safety, and neighborhood opportunities [8,9,43,49]. Hence, exposure to housing insecurity refers to conditions and situations in which individuals face challenges related to these

dimensions. Housing insecurity intersects with insecurities in other domains, such as finance, employment, family, and health as well as dimensions of inequality and discrimination such as social class, race, disability, and gender, creating a complex interplay in which these interact and mutually reinforce each other [30,50]. With regard to renters, secure housing involves finding and maintaining homes that meet tenants' needs, and is influenced by factors beyond legal tenure, including societal context (e.g., rent regulations), the market (e.g., affordability), public policy (e.g., rental assistance), cultural norms, and psychosocial dimensions of security [37,51,52]. In this systematic review, we focus on two primary dimensions of housing insecurity among renters:

*Housing unaffordability* stands out as the most extensively studied issue among renters and is the strongest standalone dimension of housing insecurity [14,37,43]. Affordability refers to the ability of renters to meet the costs of housing, including rent and other related expenses such as utilities and deposits, without experiencing excessive financial strain [8,14,43,49]. The operationalization of housing affordability varies across studies [49], with common approaches being the rent-to-income ratio (e.g., "housing-cost burden"), housing-induced poverty, residual income, and subjective measures of difficulty to afford the rent [14]. In the case of rent-to-income ratio, the threshold for when a household is considered unaffordable varies between 30 and 50% [49], sometimes with the added criterion that the household income is in the bottom 40% of the national distribution, referred to as the '30/40' measure [53].

*Housing instability* poses another primary barrier to secure housing for renters [8,14,43,49]. Stability refers to the ability of renters to remain in their housing for as long as they wish [8,14,43,49]. Common measurement approaches for housing instability among renters include eviction and frequency of moving [14]. However, compared to affordability, operationalizations of instability vary more broadly across different studies, and also include issues such as overcrowding, doubling up, duration of stay, and rental arrears [8,49].

## Objectives

The present study aimed to explore exposure to housing insecurity, i.e., unaffordability and instability, and its impact on mental health among renters. For this purpose, we focused on adults living in OECD [54] member countries to ensure greater comparability within socioeconomic contexts. All types of rental tenure (e.g., private or social renting) were included. Studies with renter samples in housing programs and subsidies directed at improving housing affordability and/or instability were also eligible for inclusion. Using a systematic review methodology and narrative synthesis, the primary objectives of this research were to comprehensively identify relevant study reports, synthesize key findings, and propose potential areas for future investigations. We further evaluated the quality of the included studies using the Joanna Briggs Institute (JBI) Critical Appraisal Tools, and the certainty of evidence using the Grading of Recommendations Assessment, Development, and Evaluation (GRADE) framework [55,56].

## Methods

The review was conducted and reported according to the Preferred Reporting Items for Systematic Reviews and Meta-Analyses [57] guidelines (PRISMA checklist provided in S1 Appendix). The review was not pre-registered and no study protocol has been published.

### Eligibility criteria

To be eligible for inclusion, studies needed to be quantitative primary or secondary research that investigated housing insecurity by examining at least one variable related to housing affordability (housing cost burden, housing affordability stress, receiving rental assistance, or living in social housing), and/or housing instability (frequent moves, forced moves, risk of eviction, behind on rent, or overcrowding). Studies investigating housing programs or policies aimed at improving affordability and/or stability (rental assistance or eviction prevention programs), were also considered for inclusion. Furthermore, studies needed to include at least one mental health-related outcome (specific mental disorders, symptoms

of psychological distress, overall mental health and well-being, psychosocial functioning, suicidal behavior, or treatment for mental health reasons, e.g., medication, outpatient, or inpatient therapy for emotional conditions). Only studies that focused on the population of adults living in rental households were eligible for inclusion, with no restriction on the type of rental tenure (e.g., private or social renting). Longitudinal studies measuring tenure status at only one time point were eligible for inclusion if they considered their population as "renters". Studies needed to be situated in OECD countries. No restrictions were made regarding publication date or status including published and unpublished studies, partially published studies, and studies from other sources like master theses, dissertations, abstracts from conference proceedings, not peer reviewed.

Studies were excluded if they analyzed other forms of housing insecurity such as housing quality, safety, neighborhood characteristics, and homelessness. Likewise, studies examining residential satisfaction and general health outcomes were excluded. Populations other than renters, such as homeowners or homeless individuals, were not eligible. While studies examining children and adolescents were also not eligible, studies that examined adult populations while including individuals who were at least 15 years old were included in the analysis. Studies may include analyses of other populations (e.g., homeowners, children) and research questions (e.g., other housing dimensions), if they provided separate analysis for our research focus. However, studies needed to specifically distinguish effects for renters, not simply adjust for tenure or include an interaction. Editorials, reviews, qualitative studies, and research with solely descriptive data were excluded.

## Search strategy

A comprehensive search of the literature was conducted in two stages by the first author M.T. under the supervision of the last author H.N. In the first stage, four electronic databases were searched: MEDLINE (PubMed), PsycINFO (EBSCO), Web of Science Core Collections, and ASSIA (ProQuest). These databases were selected because they cover a broad range of disciplines that are relevant to housing and mental health research, such as psychology, social sciences, and public health, and also feature a wide range of document types, including dissertations and conference proceedings. The search was based on three search components:

(a) the target population of renters,

(b) exposure to housing insecurity (housing affordability and/or instability), and

(c) mental health-related outcomes.

To identify relevant keywords, synonyms were searched for and key studies in the field were consulted. Additionally, Medical Subject Heading (MeSH) terms were used for MEDLINE and thesaurus index terms were used for PsycINFO and ASSIA. To further enhance the sensitivity of the search strategy, wildcard symbols were utilized and no search filters or limits were applied. The literature search was conducted on December 9, 2022. The detailed search syntax is provided in S2 Appendix.

In the second stage, citation tracking was used to further increase sensitivity and reduce potential bias due to unpublished studies or studies not indexed in the selected databases. Backward citation tracking involved screening titles and abstracts of the reference lists from included studies and related systematic reviews. Forward citation tracking was conducted from February 23–27, 2023, screening the titles and abstracts of all studies that cited the included studies.

## Study selection

After exporting all search results into the reference management program Citavi (Version 6.14), duplicate records were removed manually by the first author M.T. The initial screening process involved assessing the title and abstract of each record for eligibility. If a record was considered eligible or potentially eligible, its full text was retrieved and screened accordingly. Study selection was conducted hierarchically using a screening form, based on the research question and

predefined inclusion and exclusion criteria. Exclusion reasons for full-text reports were documented and are provided in S3 Appendix. It was ensured that studies which used the same database were distinct, as they examined separate research questions with different mental health-related outcomes and/or variables assessing housing affordability and/or instability. The screening process was conducted by the first author M.T., without the use of automation tools. In cases of uncertainty, discussions with the author H.N. were held to reach a decision. Additionally, to ensure reliability, a randomized subset of 10% of the full texts ($n = 12$) was independently screened by the author L.N. and the interrater reliability was calculated using Gwet's $AC_1$ [58] and the R package irrCAC [59].

## Data extraction and synthesis method

The study characteristics of the included studies were extracted by the first author M.T. and recorded in an extraction sheet, which comprises (a) the study characteristics (e.g., author and year, country, study design, information about the study population), (b) variables (exposure and outcome variables, as well as measurements), (c) statistical methods and study limitations and strengths, and (d) main findings. The main results were extracted separately for each variable assessing housing affordability and/or instability and mental health-related outcome measure. In the summary table, adjusted results including statistical significance, relevant effect measures (mostly odds ratios or mean differences) and confidence intervals (CI) were reported whenever possible. Unadjusted results were reported only when adjusted results were not available. In cases where data or information was missing or unclear, study authors were contacted several times by email to obtain or clarify missing data or unclear information. It was marked as not available (N/A) if it could not be obtained by contacting the study authors. After the initial extraction, a verification process was conducted by the author L.N. in a randomized subset of 30% of the included studies ($n = 7$) by checking whether the extracted information of the included studies was accurate. Discrepancies were resolved through discussions between the authors and the extracted information was clarified and corrected as needed. The interrater agreement was measured using Gwet's $AC_1$ [58] and the R package irrCAC [59].

In the evidence synthesis, the main findings of the included studies were grouped and presented by outcome and predictor variables. In an initial step, predictor variables were classified into the two dimensions housing affordability and instability. If studies examined housing programs and policies like housing benefits, social housing, or eviction moratorium, categorization was determined by their principal objectives. If studies employed different samples or multiple predictor variables belonging to different categories of housing insecurity, each of them was reported separately. Outcome measures were grouped into the categories "overall mental health, well-being, and psychological distress", "psychosocial functioning", "symptoms of mental disorders", and "mental health treatment".

## Methodological quality

The methodological quality of all included studies was assessed using the well-established JBI Critical Appraisal Tools [55,56]. These tools offer specific checklists for different study types. In this study, we used the Checklist for Analytical Cross-Sectional Studies and the Checklist for Cohort Studies [60]. The assessment was carried out by the author L.N. and 50% of the included studies ($n = 11$) were independently rated by the authors M.T. and H.N. to ensure reliability. Disagreement was resolved through discussions between the authors and the interrater agreement was measured using Gwet's $AC_1$ [58] and the R package irrCAC [59].

## Certainty of evidence

To assess the certainty of evidence regarding the effects of housing affordability and housing instability on different mental health outcomes, we applied the Grading of Recommendations Assessment, Development, and Evaluation (GRADE) instrument for observational studies [61]. First of all, we assessed the number of studies, the overall sample size, and the qualitatively summarized effect of housing affordability or housing instability on the respective mental health outcome. The

risk of bias assessment was mainly based on the JBI quality ratings. Afterwards, we rated the inconsistency, indirectness, imprecision, possible publication bias, and factors that can raise the certainty in evidence. According to the GRADE guidance, certainty of evidence of observational, non-controlled trials is always rated as low certainty, and can be upgraded or downgraded based on the individual ratings on the subscales [62]. GRADE ratings were independently assessed by the authors L.N. and M.T. Disagreements were discussed until agreement was reached. The interrater agreement was measured using Gwet's $AC_1$ [58] and the R package irrCAC [59].

### Research synthesis

Based on our knowledge of the field of research and anticipated heterogeneity in study methodologies and definitions of housing insecurity, we decided a priori that a meta-analysis would be considered if sufficient homogeneity was found—which was ultimately not the case. Therefore, a narrative synthesis was chosen over a meta-analysis to synthesize the evidence. In the results section, we will describe the characteristics and methods applied by the included studies, their methodological quality, and certainty of evidence. Further, we will report descriptive (e.g., frequencies, percentages) and inferential statistics (e.g., results from correlational analyses, regression analyses), and summarize them qualitatively.

## Results

### Study selection

The study selection process is illustrated in the PRISMA flow diagram (Fig 1). The initial search yielded a total of 1,198 potentially eligible titles (after the author M.T. had removed duplicates manually). Following title and abstract screening, 74 full-text studies were assessed for eligibility (despite multiple attempts to contact the authors, two reports could not be retrieved). Of these, 19 studies met the inclusion criteria. Based on the initial results, a subsequent step involved identifying 49 additional reports through backward and forward citation tracking. This process resulted in the inclusion of three studies. A detailed overview of the reasons for exclusion is provided in S3 Appendix. The most common reason for exclusion was that studies did not specifically analyze the association of housing insecurity and mental health among renters but rather addressed housing insecurity more broadly (e.g., statistical analyses were not separated by tenant subgroups). Overall, a total of 22 studies were included in the evidence synthesis. The interrater reliability of the study selection process was Gwet's $AC_1$ = .906, indicating high agreement [58,63].

### Study characteristics

A descriptive overview of the included studies is presented in Table 1. The interrater agreement for data extraction was Gwet's $AC_1$ = .914, indicating high agreement [58,63]. Eleven of the studies were from the U.S. [64–74], six from Australia [75–80], three from the UK [76,81,82], and one each from the Netherlands [83], South Korea [84], and Spain [50]. Fourteen studies were longitudinal studies [68–72,74–78,80,81,83,84], five were cross-sectional studies [50,64,66,67,73], and three were quasi-experimental studies without actively manipulated interventions [65,79,82]. The majority of the studies were secondary analyses. With the exception of one study [81], all were published within the last decade, mostly in 2020 or later.

The selected studies addressed a wide range of research questions, with only a few of them focusing exclusively on the population of interest. Ten out of 22 studies examined housing insecurity broadly, including not only renters but also other housing tenures such as homeowners, while analyzing renters separately in at least one analysis [65,66,72,75–78,80,81,83]. Twelve studies included only renters [64,67–71,73,74,79,82,84,85]. Six studies identified their (sub-)samples as private renters [75–78,80,82] and two as public renters [75,79]. Moreover, a number of studies examined specific population groups, such as low-income renter households [67–69,71,73,74,76–79,82; for more details see Tables 2 and 3]. It is important to note that four studies with longitudinal data only measured tenure status at one time point; therefore, it is possible that

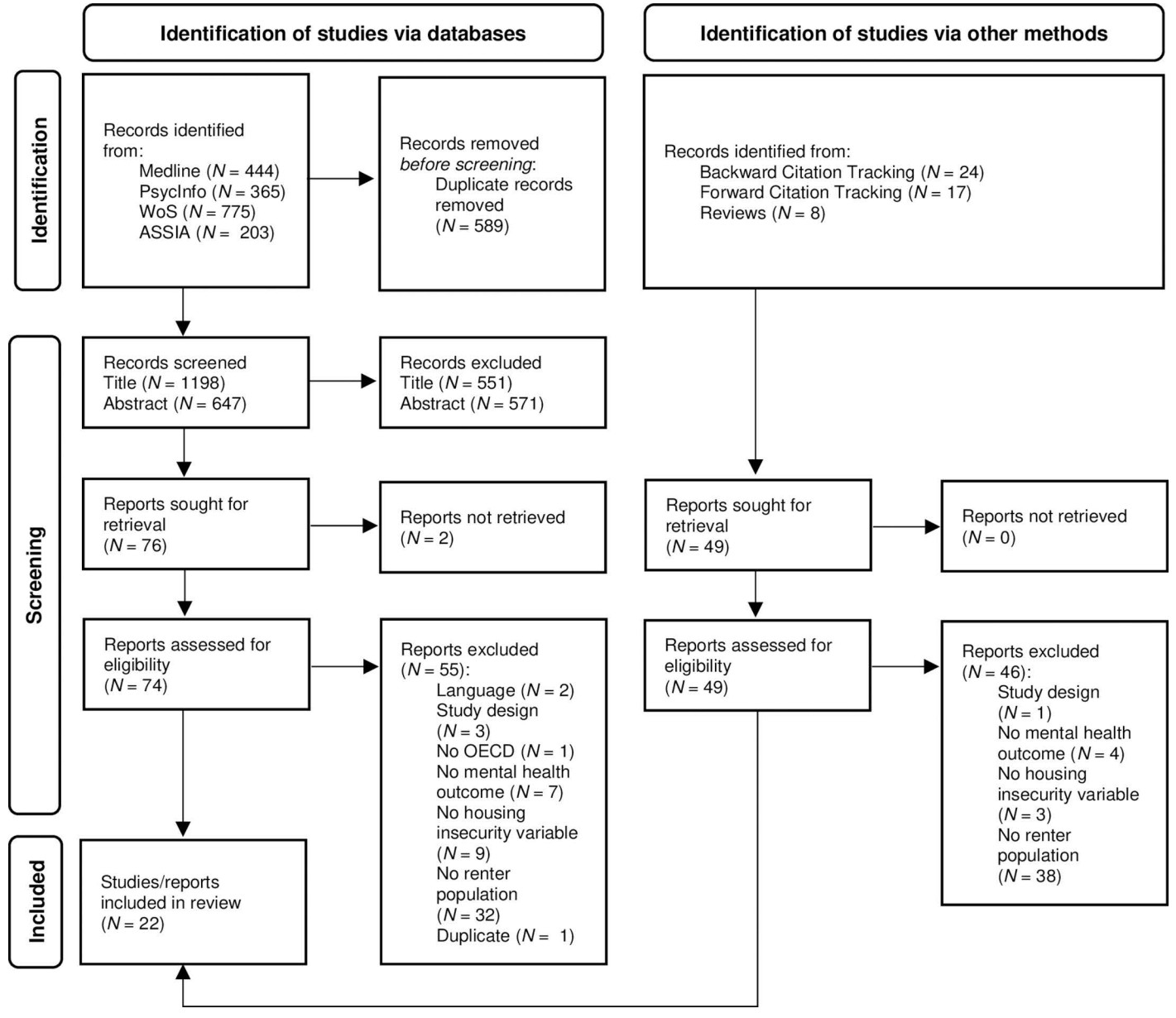

**Fig 1. PRISMA 2020 flow diagram.**

changes in tenure status occurred over time [74,79–81]. For a comprehensive overview of the characteristics as well as the main findings of the included studies, refer to Table 2 for housing affordability and Table 3 for housing instability.

### Exposure measurements of housing insecurity

Of the 22 studies included in the review, eight focused solely on housing affordability [69,72,75,76,78,79,82,83], while 13 examined housing instability [50,64–66,68,70,71,73,74,77,80,81,84; see Table 1, 2, 3], and one study investigated both dimensions [67]. Housing affordability was primarily operationalized using some form of rent-to-income ratio, while

**Table 1. Descriptive of studies included.**

| | N° (*N*=22) | % |
|---|---|---|
| **Year of Publication** | | |
| 2009 | 1 | 4.5 |
| 2010–2019 | 8 | 36.4 |
| 2020–2023 | 13 | 59.1 |
| **Country** | | |
| U.S. | 11 | 50.0 |
| Australia | 6 | 27.3 |
| UK | 3 | 13.6 |
| South Korea | 1 | 4.5 |
| Spain | 1 | 4.5 |
| The Netherlands | 1 | 4.5 |
| **Study design** | | |
| Longitudinal | 14 | 63.6 |
| Cross-sectional | 5 | 22.7 |
| Quasi-experimental | 3 | 13.6 |
| **Tenure[a]** | | |
| Renters (not further specified) | 15 | 68.2 |
| Private renters | 6 | 27.3 |
| Public renters | 2 | 9.1 |
| **Housing insecurity dimension** | | |
| Housing instability | 13 | 59.1 |
| Housing affordability | 8 | 36.4 |
| Housing affordability and instability | 1 | 4.5 |
| **Housing affordability operationalization[a]** | | |
| Rent/income ratio (e.g., housing cost burden) | 6 | 27.3 |
| Amount of 'out-of-pocket rent'/ residual income | 3 | 13.6 |
| • Amount of housing benefits | 1 | 4.5 |
| • Living in social housing | 1 | 4.5 |
| • Receiving housing assistance | 1 | 4.5 |
| **Housing instability operationalization[a]** | | |
| Forced moves due to legal or economic issues | 4 | 18.2 |
| Eviction | 3 | 13.6 |
| Frequency measure of moving | 3 | 13.6 |
| Risk of eviction | 3 | 13.6 |
| Behind on rent | 2 | 9.1 |
| Crowding/ moving in with others to save costs | 2 | 9.1 |
| Combined measure of residential instability | 1 | 4.5 |
| Duration of occupancy | 1 | 4.5 |
| **Mental health outcome[a]** | | |
| Overall mental health, well being, and psychological distress | 12 | 54.5 |
| Depression | 10 | 45.5 |
| Anxiety | 3 | 13.6 |
| Psychosocial functioning | 3 | 13.6 |
| Mental health treatment | 2 | 9.1 |
| Depression and anxiety combined | 1 | 4.5 |

*(Continued)*

**Table 1.** (Continued)

| | N° (*N* = 22) | % |
|---|---|---|
| Post-traumatic stress disorder | 1 | 4.5 |
| Problematic alcohol use | 1 | 4.5 |
| Suicidal ideation and behavior | 1 | 4.5 |

*Notes.* U.S. = United States of America; UK = United Kingdom.

ª Categories are not mutually exclusive; totals do not apply.

housing instability was mainly assessed through indicators of forced moves. With the exception of three studies that employed population-level aggregated measures [65,71,72], the remaining studies relied on individual participant and self-reported information to assess housing insecurity.

### Outcome measurement of mental health

The studies included in the review employed heterogeneous operationalizations of mental health outcomes (see Table 4). Twelve studies measured overall mental health, well-being, and/or distress, summarized here as "overall mental health" [50,65,71,74–81,83]. Three studies used measures of psychosocial functioning, such as hostility or interpersonal functioning [67,74,80]. Additionally, 11 studies focused on symptoms of mental disorders, primarily examining symptoms of depression [64,66–70,72–74,82,84]. One study investigated suicidal ideation [74]. Two studies explored changes in mental health treatment, including the prescription of medication for mental health purposes [64,74]. The majority of the selected studies utilized screening tools to measure symptoms of mental disorders (see Table 4), such as the Patient Health Questionnaire-2 [PHQ-2; 86] or screening for maternal depression [73]. Moreover, with the exception of one study in which trained clinical interviewers conducted home interviews [67] and two studies that used a clinical-diagnostic interview [68,69], all other studies relied on self-report outcome instruments.

### Methodological quality

A detailed evaluation of every item of the respective critical appraisal tool for each study is provided in S5 Appendix. Interrater agreement for the critical appraisal of the methodological quality was Gwet's $AC_1$ = .872, indicating high agreement [58,63].

In general, the five cross-sectional studies were off high methodological quality [50,64,66,67,73]. They provided inclusion and exclusion criteria, detailed descriptions of the samples, used reliable and valid measurement of outcome and, where applicable, the condition, identified and handled confounding variables adequately, and applied appropriate statistical analyses [60]. All cross-sectional studies used some form of regression analysis and controlled for multiple covariates [64,66,67,73,85], with three studies omitting income as an important potential confounder [67,73,85]. For the three studies that did not include any additional conditions beyond exposure to housing insecurity, the criterion for reliable and valid measurement of the condition (criterion 4) was rated as not applicable (N/A) [50,64,66]. Additionally, we rated self-report questions related to income or housing, as well as other measures used to assess housing insecurity and affordability, positively when widely recognized principles in housing research, such as the '30/40 rule,' were applied, even though these measures cannot be strictly tested for reliability and validity.

The quality assessment of the longitudinal [68–72,74–78,80,81,83,84] and quasi-experimental studies [65,79,82] indicated overall good quality, though some flaws were noted in specific criteria. Almost all studies measured the exposure in a valid and reliable way, identified and handled possible confounding factors adequately, and employed appropriate statistical methods. A variety of statistical models were used in the studies based on their individual study design

**Table 2. Summary characteristics of the included studies regarding housing affordability and mental health (ordered alphabetically according to author's name).**

| Study | Country | Study Design | Data Source & Years of Collection | Main Objective of Study | Study Population (Analytical Sample) | Predictor Operationalization & Measurement | Outcome Operationalization & Measurement | Main Findings | Funding |
|---|---|---|---|---|---|---|---|---|---|
| | | | | | | **Housing Insecurity** | **Mental Health** | | |
| Arundel et al. (2022) | The Netherlands | Longitudinal (Panel study) | Longitudinal Internet Studies for the Social Sciences (LISS Panel) | Examining the link between housing unaffordability and mental health and how this is differentiated across age cohorts and tenure | Renters not living in parental home (complete case analysis) | Housing affordability stress (HAS) | Mental health and well-being | Renters with HAS consistently showed lower mental health scores compared to renters without HAS, with significant differences observed at a 90% CI, except for the years 2009 & 2012. The divergence in mental health between those with and without HAS was particularly pronounced among renters in the 35—44 years age cohort. | The Dutch Research Council under Grant VI.Veni.201S.031; the Australian Research Council under Grant DP19010188; and the National Health and Medical Research Council in Australia under Grant APP1196456. |
| | | | 2008–2019 | | Age 25–65 years $N_{Renters}$ = N/A[a] | '30/40' measure[b] | MHI-5 from the SF-36 | | |
| | | | 2008-2019 | | | | | | |
| Baker et al. (2020) | Australia | Longitudinal (Panel study) | Household, Income and Labour Dynamics in Australia (HILDA) | Examining the mental health effects of (prolonged and intermittent) patterns of exposure to housing affordability problems | Private renters & public renters who took part in the survey for five consecutive waves (complete case analysis) | Housing affordability stress pattern: Pattern of exposure to HAS over time: prolonged vs. intermittent affordability problems vs. unexposed to housing affordability stress | Mental health and well-being: MCS from the SF-36 | Both the intermittent and the prolonged exposure group had lower mental health scores compared to renters unexposed to HAS, with the prolonged group showing the lowest average scores. | Data from the Household, Income and Labour Dynamics in Australia (HILDA) Survey. The HILDA Project was initiated and is funded by the Australian Government Department of Social Services (DSS) and is managed by the Melbourne Institute of Applied Economic and Social Research (Melbourne Institute). |
| | | | 2002-2016 | | Age ≥ 15 years $N_{Private}$ = 1,401[c] $N_{Public}$ = 9,694 | | | For private renters, the group mean differences between prolonged vs. unexposed and between prolonged vs. intermittent HAS were not significant. For public renters, the group mean differences were all significant (p<.01)[d]. | |

*(Continued)*

| Study | Country | Study Design | Data Source & Years of Collection | Main Objective of Study | Study Population (Analytical Sample) | Predictor Operationalization & Measurement | Outcome Operationalization & Measurement | Main Findings | Funding |
|---|---|---|---|---|---|---|---|---|---|
| | | | | | | *Housing Insecurity* | *Mental Health* | | |
| Bentley et al. (2016) [Part a] | Australia | Longitudinal (Panel study) | HILDA (2001–2008) | Examining whether the mental health of people in different tenure types was differentially affected by unaffordable housing | Low-income private renters — Age 25–64 years — $N_{Renters}$ = N/A[e] (3,036 observations) | *Housing cost burden*: Rent payments exceeded 30% of gross household income (equivalent to '30/40' measure) | *Mental health and well-being*: MCS from the SF-36 | Private renters whose housing became unaffordable experienced a small but significant decline in mental health (mean change = −0.45; 95% CI: −0.81, −0.09; $p$=.014). | Supported by the Australian Research Council (Grant Number: LP100200182 and DP120102974). |
| Bentley et al. (2016) [Part b] | UK | Longitudinal (Panel study) | British Household Panel Survey (BHPS) 2001–2008 | See [Part a] | Low-income private renters — Age 25–64 years — $N_{Renters}$ = N/A[f] (2,373 observations) | *Housing cost burden* See [Part a] | *Mental health and well-being*: GHQ-12 | Private renters whose housing became unaffordable experienced no significant change in mental health (mean change=0.30; 95% CI: −0.67, 1.26; $p$=.749). | Supported by the Australian Research Council (Grant Number: LP100200182 and DP120102974). |
| Chambers et al. (2015) [Part a] | U.S. | Cross-sectional | Affordable Housing as an Obesity Mediating Environment (AHOME) 2011-2012 | Investigating whether the type of housing assistance and the conditions of the home and neighborhood are associated with mental health outcomes of residents | Low-income Latin American renters living in the Bronx, eligible for rental assistance (complete case analysis) Age ≥ 18 years — $N$=371 | *Receiving housing assistance*: Unassisted vs. public housing residents vs. Section 8 voucher | *Symptoms of depression*: CES-D 10 / *Hostile affect*: Items of the Hostile Affect Subscale from the CMHS | Receipt of housing assistance was not significantly associated with depressive symptomology (OR = 0.857; 95% CI: 0.48, 1.53; $p$-value N/A) or hostility (OR = 1.738; 95% CI: 0.8, 3.76; $p$-value N/A). | The AHOME Study is funded by a grant from the John D. and Catherine T. MacArthur Foundation's "How Housing Matters to Families and Communities" research program (grant no. 94005–0). Dr. Chambers was also supported, in part, by National Heart, Lung, and Blood Institute research grants N01HC65235 and K01HL125466. |

*(Continued)*

| Study | Country | Study Design | Data Source & Years of Collection | Main Objective of Study | Study Population (Analytical Sample) | Predictor Operationalization & Measurement | Outcome Operationalization & Measurement | Main Findings | Funding |
|---|---|---|---|---|---|---|---|---|---|
| | | | | | | *Housing Insecurity* | *Mental Health* | | |
| Elliott et al. (2021) | U.S. | Longitudinal (Cohort study) | Fragile Families & Child Well-being Study (FFCWS) 1998–2017 | Analyzing the relationship between housing cost burden and mental health outcomes reported by a sample of rent-burdened primary caregiver mothers | Mothers/ female caregivers from low-income rental households (complete case analysis) Age N/A $N=399$ | *Housing cost burden:* Monthly rent as a proportion of income (< 31% not burdened, 31–49% burdened, ≥ 50% severely burdened) | *Symptoms of depression* Single dichotomous outcome variable for depression, constructed as responses of either "yes" or "no" as to whether or not mothers met the criteria for depression as measured by CIDI-SF items. | Mothers/female caregivers from low-income rental households who experienced rent burden (OR = 1.7, $p = .004$) or severe rent burden (OR = 1.5, $p = .005$) were significantly more likely to report liberal criteria for depression than those who did not experience rent burden. | Not reported. |
| Mason et al. (2013) | Australia | Longitudinal (Panel study) | HILDA (2001–2010) | Investigating whether a relationship exists between unaffordable housing and mental health that differs between home purchasers and private renters among low-income households | Low-income private renters who experienced a change in their affordability status during the study period Age 25–64 years $N_{Renters} = $N/A (4,362 observations)[g] | *Housing affordability stress (HAS):* '30/40' measure | *Mental health and well-being:* MCS from the SF-36 | Among low-income private renters, individuals in unaffordable housing showed a mean MCS score that was 1.18 points lower than the mean score when their housing was affordable (95% CI: 1.95, 0.41, $p = .003$). These results were consistent across different low- and mid-low-income subgroups, but were not replicated in sensitivity analyses examining higher-income households. | Funded by the Australian Research Council (Grant Number: LP100200182). T. Blakely is funded by the Health Research Council of New Zealand. |

*(Continued)*

**Table 2.** (Continued)

| Study | Country | Study Design | Data Source & Years of Collection | Main Objective of Study | Study Population (Analytical Sample) | Predictor Operationalization & Measurement | Outcome Operationalization & Measurement | Main Findings | Funding |
|---|---|---|---|---|---|---|---|---|---|
| | | | | | | **Housing Insecurity** | **Mental Health** | | |
| Prentice & Scutella (2020) | Australia | Quasi-experimental study/ longitudinal data | HILDA[h] (2001–2015) | Estimating the impacts of social housing on employment, education, health, incarceration and homelessness | Renters in social housing at time t (Control group: low-income renters)<br><br>Age ≥ 15 years<br><br>N = N/A | *Living in social housing:* Renters in community or public housing vs. low-income renters not living in social housing (matched control group) | *Psychological distress:* K6<br><br>*Mental health and well-being:* MHI-5 from the SF-36 | No significant improvements in mental health arising from social housing at t+1 (6–12 months later); indeed, social housing residents had significantly worse outcomes (ATT = 1 (MHI-5), ATT = 0.6 (K6), *p* < .05). | Funded by the Australian Government Department of Social Services (DSS); the Household, Income and Labour Dynamics in Australia (HILDA) Survey. The HILDA Project was initiated and is funded by the Australian Government Department of Social Services (DSS) and is managed by the Melbourne Institute of Applied Economic and Social Research (Melbourne Institute). |
| Reeves et al. (2016) | UK | Quasi-experimental study/ repeated cross-sectional data | Annual Population Survey (APS) 2009–2013 | Evaluating the association between housing security and mental ill health by investigating the link between a reduction in income (reduction of housing benefits) and mental health | Private renters receiving Housing Benefits (HB) (Control group: Private renters without HB)<br><br>Age 16–69 years<br><br>N = 179,037 | *Amount of housing benefits:* HB reduction in 2011 (mean reduction of approximately £1,220 per year) | *Symptoms of depression:* What health problems does the respondent currently have (list incl. "depression, bad nerves or anxiety") | The reduction of HB increased the risk of depressive symptoms among persons claiming the HB by approximately 1.8 percentage points (95% CI: 1.0, 2.7; *p* < .01) above the risk in private renters not claiming the benefit. This effect remained after matching the groups. The increase in depressive symptoms remained elevated for up to 24 months after the reform. | A. Reeves and D. Stuckler were supported by a Wellcome Trust Investigator Award. A.C. and D.S. were supported by the European Research Council (grant 313590-HRES). |

*(Continued)*

 

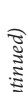

| Study | Country | Study Design | Data Source & Years of Collection | Main Objective of Study | Study Population (Analytical Sample) | Predictor Operationalization & Measurement | Outcome Operationalization & Measurement | Main Findings | Funding |
|---|---|---|---|---|---|---|---|---|---|
| | | | | | | *Housing Insecurity* | *Mental Health* | | |
| Rodgers et al. (2019) | U.S. | Longitudinal with quasi-experimental variation (Panel study) | National Longitudinal Survey of Youths 1979 (NLSY79) 2000–2014 | Examining the relationship between housing affordability and risk factors for cardiovascular disease | Renters who were 'disease-free' prior to 2000 and experienced a change in the aggregated rent burden (complete case analysis) Age 35–43 years | *Primary independent variable: Aggregated housing cost burden:* Change in the county-average proportion of total household income spent on rent (before vs. after the Great Recession in 2008) | *Change in symptoms of depression:* CES-D (before vs. after the Great Recession 2008) | Each percentage point increase in county-level median percentage of housing cost burden was associated with increased odds of depression. The association was stronger for renters than for homeowners, although the association was no longer significant when restricting the sample to renters (OR = 1.20, 95% CI: 0.92, 1.56; *p* =.17). | Supported by the National Heart, Lung, and Blood Institute at the National Institutes of Health (grant R01 HL138247; PI: D. Kim). |

$N_{Renters} = 266^i$

Notes: *CES-D 10* Center for Epidemiologic Studies – Depression Scale; *CIDI-SF* Composite International Diagnostic Interview Short-Form; *CMHS* Cook-Medley Hostility Scale; *GHQ-12* 12-item General Health Questionnaire; *K6* Abbreviated version of Kessler Psychological Distress Scale; *MCS* Mental Health Component Summary Score; *MHI-5* Mental Health Scale; *SF-36* Short Form 36 Health Survey; References for each mental health measurement test are provided in S4 Appendix.

[a] $N_{Renters} = 12,306$ (before discounting cases with missing data on key variables, final sample not specified), $N_{Total}$ = about 14,000 (1,500–1,900 per year).

[b] '30/40' measure = spending >30% of gross household income on rent and gross household income is in the bottom 40% of the national distribution [53].

[c] $N_{Total} = 48,446$.

[d] This review solely presents the descriptive findings of the study, as the regression analyses did not include stratified analyses for renters.

[e] $N_{Total} = 2,239$ (8,481 responses).

[f] $N_{Total} = 2,269$ (9,184 responses).

[g] $N_{Total} = 2,916$ (12,064 observations).

[h] The primary data source for the paper was Journey Home (JH); however, the results are not incorporated within this review, as the control group of this sample is not restricted to renters but includes all individuals not in social housing.

[i] $N_{Total} = 3,722$, of which $N_{Renters} = 1,117$; final sample size $N_{Total} = 713$ for depressive symptoms.

**Table 3. Summary characteristics of the included studies regarding housing instability and mental health (ordered alphabetically according to author's name).**

| Study | Country | Study Design | Data Source & Years of Collection | Main Objective of Study | Study Population (Analytical Sample) | Predictor Operationalization & Measurement | Outcome Operationalization & Measurement | Main Findings | Funding |
|---|---|---|---|---|---|---|---|---|---|
| | | | | | | *Housing Insecurity* | *Mental Health* | | |
| Acharya et al. (2022) | U.S. | Cross-sectional | Household Pulse Survey (HPS) 2021–2022 | Examining the association between self-reported risk of eviction and anxiety, depression, and prescription medication use for mental or emotional health reasons | Renters not caught up with rent payments at the time of the survey Age ≥ 18 years N = 14,548 | *Primary independent variable: Risk of eviction:* Self-reported likelihood of eviction in the next two months. Respondents who stated "very likely" or "somewhat likely" are considered as the at risk of eviction group | *Symptoms of depression:* PHQ-2 *Symptoms of anxiety:* GAD-2 *Usage of prescription medication for mental, emotional, or behavioral conditions:* Whether a respondent took prescription medication for mental, emotional, or behavioral health reasons | The perceived risk of eviction among renters not caught up with rent payments was associated with elevated mental health problems. The odds of depression (OR = 2.4; 95% CI: 2.36, 2.37; *p*-value N/A), anxiety (OR = 2.7; 95% CI: 2.65, 2.65; *p*-value N/A), and prescription medication use (OR = 1.2; 95% CI: 1.17, 1.17; *p*-value N/A) were significantly higher in the at-risk eviction group than in the reference group. Results from sensitivity analysis obtained after adjustment for medication use as the marker of pre-existing mental health conditions reinforced the similar magnitude of association between the risk of eviction and depression and anxiety. | No specific grant from funding agencies in the public, commercial, or not-for-profit sectors. |

*(Continued)*

**Table 3.** (Continued)

| Study | Country | Study Design | Data Source & Years of Collection | Main Objective of Study | Study Population (Analytical Sample) | Predictor Operationalization & Measurement | Outcome Operationalization & Measurement | Main Findings | Funding |
|---|---|---|---|---|---|---|---|---|---|
| | | | | | | Housing Insecurity | Mental Health | | |
| Ali & Wehby (2022) | U.S. | Quasi-experimental/ repeated cross-sectional data | Behavioral Risk Factor Surveillance System (BRFSS) 2020 | Examining the short-term effects of the state eviction moratoriums in 2020 on the mental health status of renters | Renters (Control group: home-owners to capture within-state trends) Age ≥ 18 years $N_{Renters} = 74,655$ $N_{Total} = 292,802$ | Risk of eviction. Status and duration of eviction moratorium (state-level) | Mental health and well-being: Number of days not in good mental health during the previous 30 days | State eviction moratoriums were associated with an improvement in mental health among renters, including 0.37 fewer days not in good mental health in the past 30 days ($p < .01$) and a decline in the likelihood of reporting frequent mental distress by 1.3 percentage points ($p < .05$). | Not reported. |
| Burgard et al. (2012) | U.S. | Cross-sectional | Michigan Recession and Recovery Study (MRRS) 2009–2010 | Examining the associations between different types of housing instability[a] and several measures of health | Renters (complete case analysis) | Behind on rent: "Are you currently behind on your rent?" | Symptoms of depression: PHQ-9 | Renters who were behind on rent had approximately 3.7 times greater odds of exceeding the cut-off for depression compared to renters who had not experienced recent instability (OR = 3.66; 95% CI: 1.15,11.7, $p < .05$). However, when including participants with previous housing instability ($N = 50$), those who were behind on rent were no longer more likely to exceed the cut-off for depression ($p$-value N/A). The associations between anxiety attacks (OR = 3.03; 95% CI: 0.97, 9.42, $p$-value N/A) and harmful alcohol use (OR = 2.99; 95% CI: 0.45, 20.1, $p$-value N/A) were not significant. | Data collection for this study was supported by funds provided to the National Poverty Center (NPC) by the Office of the Assistant Secretary for Planning and Evaluation at the U.S. Department of Health and Human Services, the Office of the Vice-President for Research at the University of Michigan, the John D. and Catherine T. MacArthur Foundation, and the Ford Foundation. The analyses presented in this paper were supported by a generous grant from the John D. and Catherine T. MacArthur Foundation through their How Housing Matters initiative. |

*(Continued)*

| Study | Country | Study Design | Data Source & Years of Collection | Main Objective of Study | Study Population (Analytical Sample) | Predictor Operationalization & Measurement | Outcome Operationalization & Measurement | Main Findings | Funding |
|---|---|---|---|---|---|---|---|---|---|
| | | | | | | *Housing Insecurity* | *Mental Health* | | |
| | | | | | Age 19–64 years | | *Anxiety attack:* Anxiety item from the PHQ-brief instrument | | |
| | | | | | $N_{Renters}$ = 383[b] ($n$ = 333 excluding renters with prior housing instability) | | *Problematic alcohol use:* AUDIT | | |
| Chambers et al. (2015) [Part b] | U.S. | Cross-sectional | Affordable Housing as an Obesity Mediating Environment (AHOME) 2011–2012 | See [Part a] | Low-income Latin American renters living in the Bronx, eligible for rental assistance (complete case analysis) Age ≥ 18 years | *Crowding:* Perceived household crowding (4 items created by the authors) | *Symptoms of depression:* CES-D 10 | Perceived crowding was not significantly associated with depressive symptomology (OR = 1.182; 95% CI: 0.83, 1.69; *p*-value N/A) but was significantly associated with an increased risk of hostile affect (OR = 1.821; 95% CI: 1.16, 2.85; *p* < 0.05). | The AHOME Study is funded by a grant from the John D. and Catherine T. MacArthur Foundation's "How Housing Matters to Families and Communities" research program (grant no. 94005–0). Dr. Chambers was also supported, in part, by National Heart, Lung, and Blood Institute research grants N01HC65235 and K01HL125466. |
| | | | | | $N$ = 371 | | *Hostile affect:* Items from the hostile affect subscale from the CMHS | | |

*(Continued)*

| Study | Country | Study Design | Data Source & Years of Collection | Main Objective of Study | Study Population (Analytical Sample) | Predictor Operationalization & Measurement | Outcome Operationalization & Measurement | Main Findings | Funding |
|---|---|---|---|---|---|---|---|---|---|
| | | | | | | *Housing Insecurity* | *Mental Health* | | |
| Desmond & Kimbro (2015) | U.S. | Longitudinal (Cohort study) | Fragile Families and Child Well-being Study (FFCWS) 1998–2005 | Examining the consequences of involuntary displacement from housing for low-income urban mothers | Low-income urban mothers from rental households (at baseline) who persisted through the 4th wave<br><br>Age N/A<br>*N* = 2,676 (propensity score weighting) *n* = 122 (matched models) | *Eviction:* "In the past 12 months, were you evicted from your home or apartment for not paying the rent or mortgage?"; three categories: early eviction (child age 0–1 year), midrange eviction (child age 2–3 years) and recent eviction (child age 4–5 years) | *Symptoms of depression:* CIDI-SF (during the 4th wave of the study) | Mothers from low-income rental households who experienced a recent eviction were statistically more likely to report depression compared to mothers who did not. Specifically, the predicted probability difference ranged between 0.20– 0.21 ($p$-values < .01) for both model specifications. Likewise, in the matched models the average treatment effect on the treated ranged between 0.20–0.22 ($p$-values ranging from <.1 to <.05).<br><br>The impact of eviction on maternal depression was long-lasting, enduring years after families were evicted, although its influence shrinks over time, indicated by a higher probability of depression even after early evictions ($p$ < .05). Results were robust to sensitivity analysis restricting evictions to those in which it was ensured that the exposure preceded the outcome measurement, indicated by a higher probability of depression for mothers experiencing midrange evictions ($p$ < .01). | Not reported. |

*(Continued)*

| Study | Country | Study Design | Data Source & Years of Collection | Main Objective of Study | Study Population (Analytical Sample) | Predictor Operationalization & Measurement | Outcome Operationalization & Measurement | Main Findings | Funding |
|---|---|---|---|---|---|---|---|---|---|
| | | | | | | *Housing Insecurity* | *Mental Health* | | |
| Kim & Burgard (2022) | U.S. | Longitudinal (Panel study) | MRRS 2009–2011 | Examining whether housing insecurity is associated with mental health among renters in the aftermath of the Great Recession of 2007–09. | Renters across all three waves (complete case analysis) | *Eviction:* If respondents experienced eviction (threatened, in progress, or completed) | *Anxiety attack:* Anxiety item from the PHQ-brief instrument | Respondents who had to move for cost reasons were more likely to experience anxiety attacks at follow-up by 16 percentage points ($p < .01$), with all other individual types of housing instability showing positive but non-significant associations with anxiety attacks (eviction: $p < .1$; moving in with others and multiple moves N/A). | Data collection for this study was supported by funds provided to the National Poverty Center (NPC) by the Office of the Assistant Secretary for Planning and Evaluation at the US Department of Health and Human Services, the Office of the Vice-President for Research at the University of Michigan, the John D. and Catherine T. MacArthur Foundation, and the Ford Foundation. |
| | | | | | | *Moving in with others:* If respondents had moved in with others to save on costs | *Symptoms of depression:* PHQ-9 | Respondents who experienced eviction over follow-up were more likely to meet criteria for depression at follow-up by 13 percentage points ($p < .05$). Multiple moves were marginally associated with depression ($p < .1$). All other specific types of housing instability show positive but non-significant associations with depression ($p$-values N/A). | |
| | | | | | $N = 255$ (510 observations) | *Moved for cost:* If respondents experienced cost-related move *Multiple moves:* If respondents moved more than twice over follow-up | | | |

| Study | Country | Study Design | Data Source & Years of Collection | Main Objective of Study | Study Population (Analytical Sample) | Predictor Operationalization & Measurement | Outcome Operationalization & Measurement | Main Findings | Funding |
|-------|---------|--------------|-----------------------------------|-------------------------|--------------------------------------|---------------------------------------------|-------------------------------------------|---------------|---------|
| | | | | | | *Housing Insecurity* | *Mental Health* | | |
| Leifheit et al. (2021) | U.S. | Longitudinal (Panel study) | Understanding Coronavirus in America Survey 2020 | Examining which eviction protections were associated with reduced mental distress among renters during the COVID-19 pandemic | Low-income renters (renting at any wave of the study) and who specified their state of residence Age ≥ 18 years $N$ = 2,317 (20,853 observations) | *Risk of eviction:* Time-varying strength of state-level eviction moratorium (none, weak, strong) | *Mental health and well-being:* PHQ-4 | Moratoriums that blocked landlords from giving notice for evictions (strong) were associated with a relative reduction in mental distress of 13% (risk ratio = 0.87; 95% CI: 0.76, 0.99; *p*-value N/A), whereas protections that blocked only court hearings, judgments, and enforcement (weak) did not reduce distress significantly (risk ratio = 0.96; 95% CI, 0.86, 1.06; *p*-value N/A). Results were robust to a number of sensitivity analyses. Strong moratoriums were associated with larger reductions in mental distress in states with high rental cost burden at baseline than those with lower rental cost burden. | Dr. Leifheit was supported by postdoctoral training grant T32HS000046 from the Agency for Healthcare Research and Quality. |
| Li et al. (2022) | Australia | Longitudinal (Panel study) | Household, Income and Labour Dynamics in Australia (HILDA) 2001–2019 | Examining the impact of tenure instability on mental health and psychological distress among a low-income working-age population | Low-income private renters Age 25–65 years | *Frequency of transitions:* Average number of transitions every 5 years in the current tenure *Duration of occupancy:* Number of years in the current dwelling | *Mental health and well-being:* MHI-5 from the SF-36 | More frequent transitions were significantly associated with lower levels of mental health (MHI: −1.26, 95% CI: −2.00, −0.51; rescaled K10: −0.92, 95% CI: −1.84, −0.003; *p*-value N/A) | Funding support from the National Health and Medical Research Council (NHMRC) Centre of Research Excellence in Healthy Housing (1196456). |

*(Continued)*

| Study | Country | Study Design | Data Source & Years of Collection | Main Objective of Study | Study Population (Analytical Sample) | Predictor Operationalization & Measurement | Outcome Operationalization & Measurement | Main Findings | Funding |
|---|---|---|---|---|---|---|---|---|---|
| | | | | | | **Housing Insecurity** | **Mental Health** | | |
| | | | | | $N_{Renters}$ = 3,579[c] (17,163 observations) | | *Psychological distress: K10* | Longer occupancy was significantly associated with higher levels of mental health, as indicated by positive and significant β2+ β3 (*p*-value N/A). Residential stability was particularly beneficial for renters in early middle adulthood (35–44 years). The study's sensitivity analyses to reduce residual confounding supported these findings | |
| Park & Seo (2020) | Korea | Longitudinal (Panel study) | Korean Welfare Panel (KoWePS) 2006–2019 | Examining the associations between residential instability and perceived health status of renters | Renters Age ≥ 18 years N=6,119 (28,887 observations) | *Primary independent variable: Residential instability:* Respondents had experienced overdue rent for two months or more or relocation because they could not pay their rent | *Overall Depression:* Depression index questionnaire, symptoms experienced by individuals over past week | Residential instability significantly increased overall depression level when comparing renters with and without exposure to residential instability (p < .001). | No external funding. |
| Pevalin (2009) | UK | Longitudinal (Panel study) | British Household Panel Survey (BHPS) 1991–2008 | Investigating whether repossessions and evictions increase the likelihood of common mental illness and examining patterns over time | Renters (at baseline, complete case analysis) Age N/A $N_{Renters}$ = 3,899[d] (person/ years = 22,744) | *Eviction:* Respondents who had moved since their last interview were asked why they had moved; one possible response was "eviction" | *Common mental illness:* GHQ-12 | Evicted renters exhibited elevated levels of common mental illness immediately prior to the event, but there was no significant association between the eviction event and an increased risk of mental illness after the event (OR: .97, 95% CI: .76, 1.20; p-value N/A). | No external funding. |

*(Continued)*

| Study | Country | Study Design | Data Source & Years of Collection | Main Objective of Study | Study Population (Analytical Sample) | Predictor Operationalization & Measurement | Outcome Operationalization & Measurement | Main Findings | Funding |
|---|---|---|---|---|---|---|---|---|---|
| | | | | | | **Housing Insecurity** | **Mental Health** | | |
| Sandel et al. (2018) | U.S. | Cross-sectional | Household-level surveys and medical record audits by Children's Health Watch 2009–2015 | Evaluating how housing instability relates to caregiver and child health among low-income renter households. | Caregivers of small children in low-income rental households | *Behind on Rent:* "During the last 12 months, was there a time when you were not able to pay the mortgage or rent on time?" | *Maternal depressive symptoms:* 3-item screening test developed for maternal depression | Compared with caregivers in stable housing, caregivers who were behind on rent had increased adjusted odds of maternal depressive symptoms (aOR: 2.71; 95% CI: 2.51, 2.93; *p*-value N/A). | Funded by the Blue Cross Blue Shield of Massachusetts Foundation as well as a variety of individual and foundation supporters who enabled the data collection and analysis that informed this work. The full listing is available at childrenshealthwatch.org |
| | | | | | Age N/A N = 22,324 | *Multiple moves:* More than one move in the past 12 months. | | Compared with caregivers in stable housing, caregivers with multiple moves had increased adjusted odds of maternal depressive symptoms (aOR: 3.67; 95% CI: 3.22, 4.17; *p*-value N/A). | |
| Tsai et al. (2021) | U.S. | Longitudinal (Cohort study) | 2017/2018 | Assessing housing and mental health outcomes for a cohort of eviction court participants | Renters (at baseline) with a residential eviction case at the New Haven Eviction Court in 2017/18 Age ≥ 18 years | *Residential mobility after eviction:* Participants who had to move vs. did not have to move after eviction court | *Mental health and well-being:* MCS from the SF-12v2 *Post-traumatic stress disorder:* PCL-5 *Depression and anxiety:* PHQ-4 *Psychological distress:* Subscales from the BSI[f] | There were no significant changes in mental health symptoms, suicidal ideation, or utilization of mental health treatment services between groups over time (baseline to 1, 3, 6, and 9 months; *p*-values N/A). | No specific funding. |

*(Continued)*

**Table 3.** (Continued)

**Housing Insecurity** — **Mental Health**

| Study | Country | Study Design | Data Source & Years of Collection | Main Objective of Study | Study Population (Analytical Sample) | Predictor Operationalization & Measurement | Outcome Operationalization & Measurement | Main Findings | Funding |
|---|---|---|---|---|---|---|---|---|---|
| | | | | | N = 89[e] | | *Quality of Life:* Q-LES-Q-SF<br><br>*Suicidal ideation:* If participants had suicide attempts in the past 2 years<br><br>*Receipt of any mental health or substance abuse treatment:* If participants received any treatment for mental health, alcohol, or drug problems in the past 30 days | | |
| Vasquez-Vera et al. (2022) | Spain | Cross-sectional | Survey of the Living Conditions of Renters in the Barcelona Metropolitan Area 2019 | Analyzing by gender the relationship of forced displacements due to neglected housing insecurity with the physical and mental health of renters in Barcelona in 2019 | Renters living in Barcelona city, without formal eviction | *Primary independent variables: Neglected housing insecurity:* Had to move in the last 5 years due to legal (LHI) or economic (EHI) housing insecurity | *Mental health and well-being:* Short version of the WEMWBS | The probability of worse mental health outcomes was greater in those affected by economic housing insecurity, followed by those affected by legal housing insecurity, both compared to those who have not been affected by housing insecurity. In those affected for economic reasons, this association was significant even after adjusting for socioeconomic and other housing variables, in women PR: 1.17 (95% CI: 1.03, 1.33; *p*-value N/A), in men PR: 1.21 (95% CI: 1.01, 1.43; *p*-value N/A) but not significant in those affected for legal reasons, in women PR: 1.1 (95% CI: 1.0, 1.3; *p*-value N/A), in men PR: 1.0 (95% CI: 0.8, 1.2; *p*-value N/A). | No external funding. |

*(Continued)*

| Study | Country | Study Design | Data Source & Years of Collection | Main Objective of Study | Study Population (Analytical Sample) | Predictor Operationalization & Measurement | Outcome Operationalization & Measurement | Main Findings | Funding |
|-------|---------|--------------|-----------------------------------|-------------------------|--------------------------------------|--------------------------------------------|------------------------------------------|---------------|---------|
| | | | | | | **Housing Insecurity** | **Mental Health** | | |
| | | | | | Age ≥ 18 years $N = 1,605^g$ | | | | |
| ViforJ et al. (2023) | Australia | Longitudinal (Panel study) | HILDA 2001–2018 | Examining how the links between forced housing mobility and mental well-being may vary according to states of employment and housing tenure insecurity | Private renters (at baseline) Age ≥ 15 years $N_{Renters} = 10,580$ for SF-36 (44,534 observations) $N_{Renters} = 7,413$ for K10 (16,542 observations) | *Residential mobility:* Three categories of mobility in the last year: forced housing mobility (moves due to eviction, property no longer available, living in government housing with no choice but to move, or moves made by those who had reported difficulty paying rent during the year) vs. voluntary vs. non-housing-related mobility | *Mental health and well-being:* MHI-5 from the SF-36 | Forced housing mobility reduced the mental health score by 1.2 ($p < .01$) and increased the psychological distress score by 0.6 ($p < .01$) for private renters compared to an absence of mobility (within-person effects). The odds of experiencing a high social functioning (OR: 0.8, $p < .01$) were significantly lower when a forced move occurred compared to when a move did not occur. No significant differences in the odds of experiencing a role-emotional impairment were observed (OR: 0.9, $p < .1$). | R. O. ViforJ is the recipient of an Australian Research Council (ARC) Future Fellowship (project FT200100422) funded by the Australian Government. While conducting this research, J. Hewton was the recipient of a 2020 MRes Stipend Scholarship funded by Curtin University. This research was also supported partially by the Australian Government through the ARC Discovery Projects funding scheme (project DP190101461). This article uses unit record data from the Household, Income and Labour Dynamics in Australia (HILDA) Survey. The HILDA Project was initiated and is funded by the Australian Government Department of Social Services (DSS) and is managed by the Melbourne Institute of Applied Economic and Social Research (Melbourne Institute). |

*(Continued)*

**Table 3.** (Continued)

| Study | Country | Study Design | Data Source & Years of Collection | Main Objective of Study | Study Population (Analytical Sample) | Predictor Operationalization & Measurement | Outcome Operationalization & Measurement | Main Findings | Funding |
|---|---|---|---|---|---|---|---|---|---|
| | | | | | | *Housing Insecurity* | *Mental Health* | | |
| | | | | | | | Psychological distress: K10 | | |
| | | | | | | | Role limitation, emotional: Role-emotional from the SF-36 | | |
| | | | | | | | Social functioning: Social functioning from the SF-36 | | |

*Notes: AUDIT* Alcohol Use Disorders Identification Test; *BSI* Brief Symptom Inventory; *CES-D 10* Center for Epidemiologic Studies – Depression Scale; *CIDI-SF* Composite International Diagnostic Interview Short-Form; *CMHS* Cook-Medley Hostility Scale; *GAD-2* Generalized Anxiety Disorder Scale-2; *GHQ-12* 12-item General Health Questionnaire; *K10* Kessler Psychological Distress Scale; *MCS* Mental Health Component Summary Score; *MHI-5* Mental Health Scale; *PCL-5* Posttraumatic Stress Disorder Checklist for DSM-5; *PHQ* Patient Health Questionnaire; *Q-LES-Q-SF* Quality of Life Enjoyment and Satisfaction Questionnaire-Short Form; *SF-36* Short Form 36 Health Survey; *SF-12v2* Abbreviated version of SF-36, version 2; *WEMWBS* Warwick-Edinburgh Mental Wellbeing Scale; References for each mental health measurement test are provided in S4 Appendix.

[a] The study examined eight types of housing instability; however, only the variable "behind on rent" was analyzed separately for the group of renters.

[b] $N_{Total}$ = 894.

[c] $N_{Total}$ = 7,059 (42,067 observations).

[d] $N_{Total}$ = 12,390 (139,928 person/ years).

[e] $N_{Total}$ = 121, the final analytical sample was smaller due to loss to follow-up and *N*=2 participants reporting that they moved before appearing at eviction court.

[f] The specific BSI subscale was unclear due to conflicting information. In the text, the authors mentioned the depression, anxiety, and psychoticism subscales, whereas in the tables, hostility, paranoia, and psychoticism subscales were referenced.

[g] All persons not identifying with binary gender were excluded due to a very small number of cases (*N*=32).

**Table 4. Summary of outcome operationalizations and measurement instruments used in the included studies.**

| Operationalizations | Outcome measure | | Studies |
|---|---|---|---|
| **Overall mental health, well-being, and psychological distress** | MCS summary score including the subscales vitality, social functioning, role-emotional, and mental health from the SF-36 | | *Arundel et al. (2022); Baker et al. (2020); Bentley et al. (2016); Mason et al. (2013)* |
| | MH/MHI-5 from the SF-36 | | *Li et al. (2022); Prentice & Scutella (2020); ViforJ et al. (2022)* |
| | MCS of the SF-12, abbreviated version of the SF-36 | | *Tsai et al. (2021)* |
| | GHQ-12 | | *Bentley et al. (2016); Pevalin (2009)* |
| | WEMWBS | | *Vásquez-Vera et al. (2022)* |
| | K10 | | *ViforJ et al. (2022); Li et al. (2022)* |
| | Abbreviated version of K10 (K6) | | *Prentice & Scutella (2020)* |
| | PHQ-4 | | *Leifheit et al. (2021)* |
| | Subscales of the BSI | | *Tsai et al. (2021)* |
| | Number of days in good mental health (previous 30 days) | | *Ali & Wehby (2022)* |
| | Not having good mental health for ≥ 14 days (previous 30 days) | | *Ali & Wehby (2022)* |
| **Psychosocial functioning** | Role limitation | Role-emotional from the SF-36 | *ViforJ et al. (2022)* |
| | Social functioning | Social functioning from the SF-36 | *ViforJ et al. (2022)* |
| | Quality of life/ Psychosocial status | Q-LES-Q-SF | *Tsai et al. (2021)* |
| | Hostility | Hostile affect (Ho) subscale from the CMHS | *Chambers et al. (2015)* |
| **Symptoms of mental disorders** | Depression | Depression items from the CIDI-SF | *Desmond & Kimbro (2015); Elliott et al. (2021)* |
| | | CES-D 10 | *Chambers et al. (2015); Rodgers et al. (2019)* |
| | | PHQ-9 | *Burgard et al. (2012); Kim & Burgard (2022)* |
| | | PHQ-2 | *Acharaya et al. (2022)* |
| | | Depression index questionnaire | *Park & Seo (2020)* |
| | | Current problems with depression, bad nerves, or anxiety | *Reeves et al. (2016)* |
| | | 3-item screening test for maternal depression | *Sandel et al. (2018)* |
| | Anxiety | Anxiety item from the PHQ-brief | *Burgard et al. (2012); Kim & Burgard (2022)* |
| | | GAD-2 | *Acharaya et al. (2022)* |
| | Depression & anxiety combined | PHQ-4 | *Tsai et al. (2021)* |
| | Post-traumatic stress disorder | PCL-5 | *Tsai et al. (2021)* |
| | Problematic alcohol use | AUDIT | *Burgard et al. (2012)* |
| **Mental health treatment** | Receipt of any treatment for mental health, drug, or alcohol problems | | *Tsai et al. (2021)* |
| | Usage of prescription medication for mental, emotional, or behavioral conditions | | *Acharaya et al. (2022)* |

*(Continued)*

**Table 4.** (Continued)

| Operationalizations | Outcome measure | Studies |
|---|---|---|
| **Suicidal ideation and behavior** | Suicidal ideation | *Tsai et al. (2021)* |

Notes: *AUDIT* Alcohol Use Disorders Identification Test; *BSI* Brief Symptom Inventory; *CES-D 10* Center for Epidemiologic Studies – Depression Scale; *CIDI-SF* Composite International Diagnostic Interview Short-Form; *CMHS* Cook-Medley Hostility Scale; *GAD-2* Generalized Anxiety Disorder Scale-2; *GHQ-12* 12-item General Health Questionnaire; *K10* Kessler Psychological Distress Scale; *K6* Abbreviated version of Kessler Psychological Distress Scale; *MCS* Mental Health Component Summary Score; *MHI-5* Mental Health Scale; *PCL-5* Posttraumatic Stress Disorder Checklist for DSM-5; *PHQ* Patient Health Questionnaire; *Q-LES-Q-SF* Quality of Life Enjoyment and Satisfaction Questionnaire-Short Form; *SF-36* Short Form 36 Health Survey; *SF-12v2* Abbreviated version of SF-36, version 2; *WEMWBS* Warwick-Edinburgh Mental Wellbeing Scale; References for each mental health measurement test are provided in S4 Appendix.

and dataset, with the majority of them utilizing robust methods to adjust for confounders such as multivariate regression, matching, and propensity score matching. For instance, most longitudinal studies used some form of regression analysis with fixed effect, enabling within-person comparisons over time and controlling for time-invariant individual characteristics [69,71,72,76–78,80,81,84]. Li et al. [77] employed additionally a marginal structural model addressing time-varying confounding variables and selection biases. Several studies combined regression analysis with matching methods, to achieve covariate balance between exposed and unexposed groups [68,70,79,82]. Reeves et al. [82] used a regression model that included a difference-in-difference design and Ali and Wehby [65] a triple-difference design, offering control over potentially unobserved confounding factors by comparing treatment and control groups before and after the intervention. Notably, one longitudinal study solely conducted bivariate analyses without accounting for potential confounders, which indicates a significant risk of bias [74]. Arundel et al. [83] did not specify the statistical methods utilized at all.

For criterion 1, the JBI requires no significant differences between the control and intervention groups on any potentially relevant variable at baseline. Almost all included studies clearly defined inclusion and exclusion criteria and recruited both groups from the same population (i.e., renters) to ensure that the two groups be similar with comparable characteristics except for their exposure status. Since some groups differed by definition (i.e., in terms of income), almost none of the studies fulfilled this criterion. We would like to highlight that this seems to be based on the specific study designs included in this review rather than from issues of methodological quality per se. Besides, this issue was mitigated by the fact that all studies performed confounder analyses and/or focused on within-person changes, reducing the impact of differences between the groups.

Criterion 6 – which requires that the groups be free of the outcome at baseline – was not fulfilled by many of the included studies, as both the intervention and control groups often exhibited mental health issues from the outset, which is to be expected in the general population [87]. However, this poses a potential source of bias, as the presence of pre-existing mental health conditions may influence the outcomes and confound the effects of the exposure, making it more difficult to isolate the true impact of the studied variables. At the same time, the majority of longitudinal studies used a within-subject design, which helps mitigate this potential bias by focusing on changes over time within the same individuals, rather than between groups [65,72,74,76,78,84]. Additionally, many studies measured mental health and psychological symptoms on a continuum rather than as categorical diagnoses, allowing statistical control of effects.

Beyond that, one frequently omitted aspect in the method section of both cross-sectional and longitudinal studies was the numerical reporting of psychometric properties of the outcome instruments. Some studies did not report any psychometric properties [65,82,84] while others provided at least references to validation studies [64,66,68–76,78–81,83,85]. Additionally, some studies did not address the psychometric properties of the measurements of housing insecurity at all [64,66,71]. Furthermore, since the majority of studies included in this review were based on secondary analyses of survey data, some specific aspects need to be considered. While all studies reported the source and most important characteristics of the original data set, some did not provide comprehensive information on missing data, follow-up procedure, as well

as loss to follow-up [69,71,72,74,76,79–81,83,84; for the longitudinal studies]. This lack of information contains a potential source of bias.

In summary, the majority of the included studies employed robust study designs and statistical analyses that addressed several of the methodological shortcomings mentioned above and the resulting risk of bias. Due to the research questions and impracticality of rigorously controlled and randomized studies investigating housing insecurity and affordability, most of the included studies minimized the risk of bias (for details see S5 Appendix).

## Certainty of evidence

A detailed evaluation of every item of the GRADE rating for each mental health outcome is provided in S6 Appendix. Interrater agreement was Gwet's $AC_1 > .999$, indicating high agreement [58,63]. In total, the certainty of evidence for the 12 mental health outcomes was rated as low or very low, respectively. This overall low to very low certainty of evidence is largely a result of the study designs rather than poor quality per se. In the GRADE framework, non-randomized, observational studies are initially rated as low certainty and can only be upgraded under specific circumstances [62].

Studies on housing affordability investigated overall mental health (five studies), depression (four studies), and psychosocial functioning (one study). Overall mental health and depression were rated as low certainty. Despite small numbers of studies for these outcomes, sample sizes of both outcomes were high, and populations and measures were diverse – which seems reasonable in the context of housing insecurity. Still, the majority of studies pointed in the same direction with similar effects. Only one study investigated the effect of housing affordability on psychosocial functioning. Although the study is of high methodological quality, the sample size is small and the population under study very specific. Therefore, we rated the overall certainty of evidence as very low.

Studies on housing instability investigated overall mental health (seven studies), depression (seven studies), anxiety (three studies), psychosocial functioning (two studies), mental health treatment use (two study), depression and anxiety combined (one study), posttraumatic stress disorder (one study), problematic alcohol use (one study), and suicidal ideation (one study). We rated the certainty of evidence regarding overall mental health as low. Only two studies were of high methodological quality, but the overall sample size was large, and effects were generally consistent across populations and operationalizations of housing instability. Effects mostly point in the same direction of a small negative effect of housing instability on overall mental health with little variability (i.e., small CIs). For depressive symptoms in the context of housing instability, we also rated the certainty of evidence as low. Six out of seven studies show high methodological quality and the other aspects of the rating show no or borderline concerns. Most of the studies reported small negative effects of housing instability on depressive symptoms. Still, since no factors contribute to formally upgrade the level of certainty of evidence, it remains a low certainty rating. The three studies addressing housing instability and anxiety show different study designs and mixed results. We rated the certainty of evidence as low. All three studies are of high methodological quality. At the same time, they applied different methodological approaches in different populations, making a direct comparison and aggregation of their findings difficult. No factors contributed to an upgrading or downgrading of the certainty rating. We rated the certainty of evidence of the included studies investigating housing instability and psychosocial functioning as low. Only two studies investigated psychosocial functioning as mental health outcome but with a large total sample size. The populations under study and methods applied are heterogeneous. Nevertheless, for two out of three outcome measures a small negative effect of housing instability on psychosocial functioning was reported. The studies controlled adequately for confounding factors, but only one study reported (moderate) confidence intervals. Two studies addressed the relation between housing instability and mental health treatment use with an overall very low certainty of evidence. Only one study showed a high methodological quality. We assessed (very) serious concerns regarding study quality, inconsistency, and borderline concerns regarding imprecision which led to the downgrading of the certainty of evidence of the outcome mental health treatment use. All other outcome measures in the context of housing instability (depression and anxiety combined, posttraumatic stress disorder, problematic alcohol use, and suicidal ideation)

have been studied by only one study each and no significant effect has been reported. Depression and anxiety combined, posttraumatic stress disorder, and suicidal ideation have been part of the same longitudinal study with small sample size (N = 89) and moderate methodological quality. Therefore, the evidence of the three outcomes shows the same methodological limitations which led to the rating of very low certainty of evidence in all three cases. Problematic alcohol use has been studied by one cross-sectional study. We rated (very) serious concerns regarding study quality, imprecision, and publication bias leading to an overall rating of very low certainty of evidence regarding housing instability and problematic alcohol use (for details see S6 Appendix).

## Study results

### Housing affordability and mental health

Of the nine studies examined in this review that investigated the association between housing affordability and mental health among renters, six found that unaffordable housing was significantly associated with a decline in mental health. Five of these studies employed a longitudinal design [69,75,76,78,83] and one a quasi-experimental design [82]. One quasi-experimental study [79] reported an inverse association, while one cross-sectional [67] and one longitudinal study [72] found no significant association between housing affordability and mental health.

The majority of the studies (eight out of nine) on affordability focused on the low-income population, either directly, by limiting their sample to low-income households, or indirectly, by utilizing the '30/40' measure. Only one study, conducted by Rodgers et al. [72], analyzed the general population of renters, while another study, by Mason et al. [78], conducted an additional analysis investigating high-income renters; both studies found no association. Among the findings specific to low-income renters, six showed statistical significance [69,75,78,82,83,88], whereas two yielded statistically non-significant results [67,79].

When considering only those studies that measured overall mental health, the majority (four out of five) demonstrated a significant association between unaffordable housing and worse overall mental health. Specifically, three studies found that high rental burden had a negative impact on the overall mental health of both Australian public renters [75] and private renters [76,78]. A similar trend was confirmed by Arundel et al. [83] for the Netherlands. Two of the studies reported no significant association for private renters in the UK [76] and in Australia [75]. Prentice and Scutella [79] found an inverse association, indicating that those living in social housing had significantly worse overall mental health than the matched control group in the private rental sector.

Four studies explored the association between housing affordability and depressive symptoms among renters. Elliott et al. [69] found a significant positive association between rent burden and maternal depression. Rodgers et al. [72] found a positive association between aggregated rent burden and depressive symptoms in a sample across tenures, which was not statistically significant when only the smaller subsample of renters was analyzed. Reeves et al. [82] observed a significant increase in renters' symptoms of depression after the reduction of housing benefits. In a sample of low-income Latin American renters, neither depressive symptoms nor hostile affect were significantly associated with receiving rental assistance [67].

### Housing instability and mental health

Of the 14 studies that examined the link between housing instability and mental health, 12 reported one or more significant associations between unstable housing and poorer mental health among renters. Six of these studies employed a longitudinal design [68,70,71,77,80,84], five a cross-sectional design [64,66,67,73,85], and one a quasi-experimental design [65]. Two longitudinal studies found no significant associations [74,81].

When considering only those studies that specifically measured overall mental health, five out of seven reported a significant association between unstable housing and poorer mental health. Specifically, Leifheit et al. [71] and Ali and Wehby [65] observed a positive association between the eviction moratorium implemented in the U.S. in response to the

COVID-19 pandemic and improved mental health of renters. Additionally, three studies found that experiencing forced mobility [80,85], or a higher frequency of moves [77], respectively, were associated with a negative impact on the mental health of private renters in Australia and Spain. Conversely, longer duration in the rental property was linked to improved mental health [77]. Two studies which focused on eviction [81] and forced mobility after eviction [74] did not find any statistically significant association with overall mental health.

When examining the studies that investigated depressive symptoms as outcome measures, the majority (seven out of eight) found a significant association between unstable housing and a rise in depressive symptoms. Indicators including (risk of) eviction, forced moves, multiple moves, and being behind on rent showed a link to increased depressive symptoms [64,66–68,70,73,84]. However, two studies, which examined subjective crowding [67] and moving in with others as well as forced mobility [70], did not find a significant association with depressive symptoms.

Three studies examining symptoms of anxiety were included in the review. The results revealed that subjective risk of eviction [64] and forced mobility [70] were significantly associated with greater anxiety symptoms. The other measures examined in the study by Kim and Burgard [70], including eviction, multiple moves, and moving in with others, as well as being behind on rent in the study by Burgard et al. [66], were not found to be significantly associated with anxiety.

Furthermore, the review included one study on symptoms of post-traumatic stress disorder [74], one on alcohol abuse [66], and one on a combined measure of depression and anxiety [74]. Tsai et al. [74] was the only study to investigate suicidal ideation. None of these studies found a significant association between unstable housing and these mental health outcomes. However, one study reported a higher prevalence of prescription medication for mental health reasons [64], while mental health treatment as an outcome measure was not found to be significant in the other study [74]. Finally, two studies reported a significant association between unstable housing and impairments in social and interpersonal functioning [67,80].

## Discussion

### Overall findings

This systematic review aimed to examine the association between housing insecurity and mental health among renters, with a specific focus on exposure to unaffordability and instability as key dimensions of housing insecurity. Through a comprehensive search process, a total of 22 studies that met the inclusion criteria were included.

### Housing affordability

Among the nine studies examining housing affordability [67,69,72,75,76,78,79,82,83], six reported significant associations between unaffordable rent, more specifically rent burden and reduction in housing benefits, and low mental health [69,72,75,78,82,83]. The majority of these studies (seven out of nine) focused solely on the low-income renter population. Thus, current evidence points to an association between unaffordable rent and poor mental health for this population. Notably, the studies with significant findings all took income into account, mitigating the risk of conflating the effect of (low) income and unaffordable rent on mental health [75]. In addition, previous publications also suggest that the burden of unaffordable housing mainly affects low-income renter and homeowner households, whereas high-income households do not experience such effects [89,90]. Renters with economic capital are likely to face fewer challenges related to housing insecurity, as they possess various means to compensate for these issues [14,90]. Moreover, the synthesis indicates that the association between unaffordable housing and mental health might differ depending on the context, in particular being less pronounced in countries with strong tenure protections. This pattern was observed in studies comparing different countries [76] or exploring the consequences of deterioration of rental policies [82,83]. These findings are consistent with other research investigating housing across different countries, although they are not restricted to the population of renters [34,36,51].

However, in the current review, the two included studies that examined housing affordability interventions (social housing and housing assistance) did not reveal differences in mental health outcomes compared to either no intervention [67] or longitudinally after 6–12 months [79]. Recent systematic reviews encompassing households across different tenure statuses have also reported inconsistent evidence regarding the association of interventions promoting affordability and stability with improved mental health outcomes [48,91]. Therefore, it remains unclear which specific interventions effectively enhance mental health among renters, highlighting the need for further studies in this area.

Our conclusions are mainly based on studies with overall good methodological quality (rated with the JBI appraisal tools), but overall low certainty of evidence (rated with GRADE). Conclusions must therefore be interpreted with caution. However, the low GRADE rating is partly a result of the conservative nature of the GRADE framework, which classifies observational studies—common and often necessary in this field—as low certainty by default. While this conservative approach supports robust conclusions in clinical research based on randomized controlled trials [compare 92], it may undervalue the strength of evidence in contexts like housing instability, where experimental designs are rarely feasible. Many of the included studies employed longitudinal or prospective designs, or were based on large, nationally representative datasets, which supports their findings despite some methodological limitations. Further, methodological quality of many included studies was rated as overall good when using JBI critical appraisal tools for the respective study design. In general, many studies on housing unaffordability—both included and not included in the current review—point in the same direction. Nonetheless, to more firmly establish this relationship and particularly to assess the impact of targeted interventions, future studies with rigorous and transparent methodological approaches are needed. While current findings offer important insights, conclusions should remain tentative until further high-quality evidence is available.

### Housing instability

Among the 14 studies examining housing instability [50,64–68,70,71,73,74,77,80,81,84], 12 reported significant associations between unstable housing and poor mental health among renters [50,64–66,68,70,71,73,77,80,81,84]. The significant associations were established through a diverse range of measures of housing instability, most commonly the risk of eviction, forced mobility, and frequency of transitions. Several studies also demonstrated positive effects of stable rental tenure and state eviction moratoriums in mitigating these adverse impacts. The absence of a standard definition for housing instability is evident from the diverse measures employed in the selected studies. However, the fact that many of these measures demonstrated a significant association with renters' well-being emphasizes the importance of including a range of exposure measures in a unified definition of housing instability [14,49]. This will enable future studies to delve more deeply into the multiple pathways through which housing instability appears to affect renters' well-being.

Again, the evidence examining housing instability and mental health outcomes were both rated with overall good methodological quality (based on JBI ratings) and (very) low certainty, especially for the outcomes based on only one study (based on GRADE ratings). This partly limits the validity of our conclusions. Nevertheless, the downgrading of the certainty of evidence was mainly based on the non-randomized, non-controlled study designs of included studies. Still, several studies show moderate to high methodological quality and some studies with large sample sizes point into the same direction of a negative effect. Certainty of evidence for housing instability and depression seems to be the most robust evidence in the current systematic review. Six out of seven studies show high methodological quality and the other aspects of the rating show no or borderline concerns. Most of the studies reported small negative effects of housing instability on depressive symptoms. Still, since no factors contribute to formally upgrade the level of certainty of evidence, it remains a low certainty rating. Thus, although findings and conclusions need to be interpreted against the background of general low certainty of evidence, a negative effect of housing instability and mental health is more likely than no effect.

## Implications for research

Housing insecurity often manifests across multiple dimensions simultaneously, such as affordability, (in-)stability, and quality, and intersects with further social and economic disadvantages such as gender, social class, disability, and race [9,30,50,90,93–95]. On a micro-level, these factors can accumulate and reinforce each other, amplifying their adverse effects on mental health. For instance, in a review examining gender-based inequalities in the effects of housing on health, findings suggest that while negative impacts are observed across all genders regarding housing affordability, instability, and insecurity, women, non-binary, and transgender people tend to experience more pronounced effects [50]. However, the studies included in our analysis exclusively examined renters as a whole. Recognizing the heterogeneity within this group and the likelihood of varied experiences of housing insecurity and its impact on mental health based on economic and social circumstances, it is essential for future research to adopt an intersectional lens and a bidirectional approach [cf. conceptual framework proposed by Vásquez-Vera et al., 50]. Studies further indicate that housing insecurity and its impact on mental health also varies on a macro-level, across different social, legal, and cultural contexts [34,36,37,50,51,76]. Most of the included studies (19 out of 22) in this review were conducted in the UK, the U.S., and Australia [64–82], underlining the need for research from a broader range of countries. This is particularly relevant as housing insecurity is becoming increasingly prevalent, affecting the mental health of renters even in countries with previously good tenure security such as the Netherlands [83]. Future research may additionally incorporate clinical interviews as well as measures that specifically capture other mental health problems potentially linked to housing insecurity, such as anxiety, suicide, and substance abuse [4]. In general, more robust and strong evidence is needed to substantiate our knowledge on housing insecurity and mental health. As randomized controlled trials are often not feasible in this field of study, innovative research designs, advanced statistical approaches, and triangulation of different research approaches may enable more reliable conclusions in the future.

## Implications for practice

From a clinical perspective, the findings highlight the importance of considering social and structural factors in understanding and effectively treating mental health issues. Health professionals could integrate screening questions for housing insecurity into the assessment, and remain aware of its potential impact throughout treatment, given its potential role as a triggering, maintaining, or reinforcing factor. More generally, adopting community psychological approaches that integrate mental healthcare with social work and legal support might be beneficial. The use of a multidisciplinary approach could include helping tenants who are experiencing housing insecurity to find affordable housing, access housing assistance, or navigate or resolve legal issues. Similar approaches have been successfully employed in homelessness prevention programs that align with the aim of achieving "zero discharge into homelessness" [96,97]. As part of a multidisciplinary approach, health professionals would have a unique opportunity to shape public discourse, drive research efforts, and inform policy discussions to advocate for transformative change in the conditions contributing to social disadvantages [3]. At the same time, our findings and the current literature on the topic imply the critical need to enhance access to affordable and stable housing options for renters, especially low-income households, as a means to combat housing insecurity and enhance public mental health. Structural factors such as welfare state policies, housing systems, and market dynamics significantly influence housing insecurity and insufficient public expenditure on housing exacerbates this issue [41,42,47]. Therefore, prioritizing policy responses to address housing insecurity is vital for promoting the well-being and resilience of affected individuals and communities. However, identifying effective housing policy measures, particularly for vulnerable renter groups such as low-income, disabled, or racially marginalized households, requires further research and investigation [9].

These conclusions are mainly based on evidence with acceptable and good methodological quality (based on JBI ratings), whose certainty of evidence has been rated as (very) low (using GRADE ratings). Nevertheless, single large,

longitudinal studies with robust methodology point into the same direction of negative associations between housing insecurity and mental health. Further, our understanding of individual factors contributing to mental health problems is much more advanced than our knowledge regarding structural determinants of mental health and from a less individualized point of view. Thus, more research and development of robust study designs and statistical approaches are necessary, to improve certainty of evidence in this field of research.

## Strength of evidence

The findings of this review contribute to the growing body of research on the social determinants of mental health [1–8]. The review provides aggregated evidence to support an association between housing insecurity and impaired mental health, with a substantial number of studies addressing each dimension, i.e., housing affordability and instability. In the case of housing affordability, the available data primarily concerned samples of low-income households, limiting the scope of conclusions to this specific population group. A significant proportion of the studies employed robust study designs, such as longitudinal [n = 15; 68–72,74–78,80,81,83,84] and quasi-experimental approaches [n = 3; 65,79,82], and the JBI ratings of the methodological quality of the included studies were overall good. Still, the overall GRADE rating of certainty of evidence was (very) low, mainly due to the non-randomized study designs of the included studies. In general, the majority of studies demonstrated significant associations between measures of housing insecurity for both housing dimensions and overall mental health and depressive symptoms – which is in line with other research on housing insecurity and mental health in different populations.

In contrast, the evidence for other mental health outcomes such as anxiety, mental health behavior, mental health treatment, suicidal ideation or behavior, and psychosocial functioning was more limited and inconsistent. For most of these outcomes, certainty of evidence was rated as very low using the GRADE framework, and the included studies showed some methodological limitations assessed by JBI ratings. Due to the paucity of studies, further high-quality research is necessary to draw definitive conclusions about the associations of these outcomes with housing insecurity. Moreover, most of the outcome measurement tools used in the included studies were screening tools and relied on self-report data, and thus did not provide a clinical diagnosis or a comprehensive assessment of mental disorders, although most have shown good reliability and validity, making them useful for monitoring general mental health [98]. However, two studies relied solely on single-item questions [65,82], while another two studies did not provide sufficient detail for an assessment of psychometric properties [73,84].

Despite a noticeable shift in recent years, the number of publications specifically addressing housing insecurity among renters remains limited. Furthermore, the sample sizes were often small due to secondary analyses, leading to low statistical power and hindering stratified analysis. Moreover, the included studies that yielded non-significant findings might also have been underpowered, and an effect may potentially have become detectable with sufficient testing strength [see for example [72].

Although most findings show associations between unaffordable and unstable housing and mental health in different populations and contexts, methodological limitations prevent the drawing of causal conclusions. Housing insecurity is intertwined with various other variables that might contribute to mental health problems, including socioeconomic factors, stress and negative life events, neighborhood disadvantages, and other social inequality aspects. In addition, the relationship between housing insecurity and mental health is likely to be bidirectional. Previous research across different tenure statuses suggests that pre-existing mental health conditions increase the risk of housing insecurity outcomes, while housing insecurity also negatively affects mental health [35,99,100]. The included cross-sectional studies, in particular, are susceptible to biases due to the lack of temporality, confounding, selection bias, and endogeneity, which hinder causal interpretation. However, this review incorporates also numerous studies utilizing robust methodologies like longitudinal studies with regression analysis with fixed effects, marginal structural models, or quasi-experimental designs for both dimensions of housing insecurity. In the absence of randomized controlled trials, these approaches improve causal

inference by addressing time-varying and invariant confounders and establishing temporal precedence. Nevertheless, the possibility of unobserved confounders and reverse causation remains which is reflected in the low ratings of certainty of evidence.

## Limitations

It is important to acknowledge some limitations of this systematic review. First, only a random 10% of the studies underwent double screening, and similarly, data extraction was independently verified for 30% and quality assessment for 50% of the included studies. Only in the case of the GRADE rating for certainty of evidence, 100% were independently conducted by two raters. This may introduce bias and increase the risk of missed studies. Second, this review focused solely on two dimensions of housing insecurity, i.e., affordability and instability, even though housing insecurity is a multidimensional concept that includes further challenges such as housing conditions, safety, and neighborhood opportunities. Third, we did not specifically explore the mechanisms linking housing insecurity and mental health, which could include factors such as stress, neighborhood conditions, disrupted routines, social integration, social support, stigma, financial stress, and access to healthcare [3,4,7,8,101]. Fourth, there was considerable heterogeneity between the designs of the individual studies, making it difficult to directly compare the results and to conduct a meta-analysis. Specifically, the operationalizations of housing affordability and instability, as well as mental health outcomes, varied significantly across the included studies (see Tables 1 and 4). This is also reflected in our GRADE rating of certainty of evidence. However, due to the limited number of studies, discerning which housing insecurity factors and/or mental health outcomes are more or less relevant remains challenging. Confining the analysis to particular variables and operationalizations would have significantly reduced the pool of studies available for analysis. A potential avenue for future research could involve broadening the inclusion criteria to encompass studies that do not differentiate tenure or expanding the range of included studies by including physical health outcomes. Fifth, although we did not exclude studies based on language, given that our keywords were restricted to English, it is possible that the actual number of papers on this topic is higher and that we might have overlooked valuable studies in other languages. Lastly, the eligibility criteria for this review were restricted to OECD countries, limiting the generalizability of the findings. However, some literature from non-OECD countries also points in a similar direction [34,102,103].

## Conclusion

This systematic review is the first to specifically investigate the association between housing insecurity and mental health among renters, addressing a crucial gap in the literature. The findings indicate an association of renters' exposure to housing unaffordability and instability with adverse mental health outcomes, including overall mental health and depressive symptoms. Based on the current findings, health professionals might consider housing insecurity as a contributing and aggravating factor regarding mental health issues. Housing insecurity poses a global challenge for renters in OECD countries, highlighting the need for policymakers to implement supportive housing policies and tenure protection measures in order to improve renters' housing security and ultimately public health.

## Supporting information

**S1 Appendix. Preferred Reporting Items for Systematic Reviews and Meta-Analyses (PRISMA) Checklist 2020.**
(DOCX)

**S2 Appendix. Complete Search Syntax.**
(DOCX)

**S3 Appendix. Screening Form (Full-Text) Including Reasons for Exclusion for Each Study Report.**
(XLSX)

**S4 Appendix. Reference List of the Measurement Tools Used in the Included Studies.**
(DOCX)

**S5 Appendix. Methodological Quality of the Included Studies.**
(XLSX)

**S6 Appendix. GRADE Rating of the Different Outcome Measures.**
(XLSX)

## Acknowledgments

We would like to thank Sarah Mannion for the language editing of the manuscript.

## Author contributions

**Conceptualization:** Mira Talmatzky.

**Investigation:** Mira Talmatzky, Laura Nohr.

**Methodology:** Mira Talmatzky, Laura Nohr.

**Project administration:** Mira Talmatzky, Helen Niemeyer.

**Supervision:** Helen Niemeyer.

**Validation:** Laura Nohr.

**Visualization:** Mira Talmatzky.

**Writing – original draft:** Mira Talmatzky.

**Writing – review & editing:** Laura Nohr, Christine Knaevelsrud, Helen Niemeyer.

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
