## [Decision Letter · Decision Letter 0]

8 Feb 2024

PONE-D-23-35202Exploring the association between housing insecurity and mental health among renters: A systematic reviewPLOS ONE

Dear Dr. Talmatzky,

Thank you for submitting your manuscript to PLOS ONE. After careful consideration, we feel that it has merit but does not fully meet PLOS ONE’s publication criteria as it currently stands. Therefore, we invite you to submit a revised version of the manuscript that addresses the points raised during the review process.

The reviewers have provided helpful information regarding an assortment of things concerning your manuscript.  A consistent concern among both reviewers was the need to expand your lit review sources.  Still, it appears that both reviewers feel the manuscript deserves some consideration once these edits are made.

We look forward to receiving your revised manuscript.

Kind regards,

Andrew T. Carswell

Academic Editor

PLOS ONE

Reviewers' comments:

Reviewer's Responses to Questions

**Comments to the Author**

1. Is the manuscript technically sound, and do the data support the conclusions?

Reviewer #1: Yes

Reviewer #2: Yes

2. Has the statistical analysis been performed appropriately and rigorously? 

Reviewer #1: Yes

Reviewer #2: N/A

3. Have the authors made all data underlying the findings in their manuscript fully available?

Reviewer #1: No

Reviewer #2: Yes

4. Is the manuscript presented in an intelligible fashion and written in standard English?

Reviewer #1: Yes

Reviewer #2: Yes

5. Review Comments to the Author

Reviewer #1: This article is very interesting. I make some comments on it:

Main comments

1.Introduction and discussion: To understand why access to housing is a problem nowadays it is important to present a conceptual framework of the causes, eg: public housing policies and the market. Many countries have left the access to housing to the market and therefore prices have increased and the accessibility has decreased. Moreover public spending on housing is very low. The paper should mention that. See for example:

-Marí-Dell'Olmo M, Novoa AM, Camprubí L, Peralta A, Vásquez-Vera H, Bosch J, Amat J, Díaz F, Palència L, Mehdipanah R, Rodríguez-Sanz M, Malmusi D, Borrell C. Housing Policies and Health Inequalities. Int J Health Serv. 2017 Apr;47(2):207-232. doi: 10.1177/0020731416684292. Epub 2016 Dec 28. PMID: 28030990.

2. Introduction, results and discussion: Housing insecurity change by the axes of inequality (gender, social class, race…). It is necessary to deeper emphasize this reality. Do the papers found analyse this fact? See for example:

-Vásquez-Vera C, Fernández A, Borrell C. Gender-based inequalities in the effects of housing on health: A critical review. SSM Popul Health. 2022 Mar 11;17:101068. doi: 10.1016/j.ssmph.2022.101068. PMID: 35360438; PMCID: PMC8961216.

3. Results: I am not sure that tables 1 and 2 have to be inside the paper, they can be in an appendix. In the results section I would include a table showing the descriptive results found: the number of papers for each variable (including the mental health results).

4. Discussion: an important limitation of the paper is the language chosen by the review. They exist articles in other languages that are lost. The article should mention this limitation.

Other minor comments:

Methods: please explain the meaning of “data points”. Does it mean cross sectional data?

Explain the meaning of “PTSD”.

Carme Borrell

Reviewer #2: This is an important topic with a growing literature, and it is natural to have a review piece that synthesizes the increasing evidence for the relationship between housing insecurity and mental health. The authors identify 22 studies examining the relationship between some measure of housing affordability or stability and mental health. There are several strengths of the review. First, the authors consider multiple dimensions of housing insecurity. The authors also make some attempt to assess the evidence quality or describe the study methodology. Although I think the motivation for a review piece on this topic is clear, I have some concerns about the approach used, and am somewhat skeptical the authors have identified an exhaustive list of studies.

Main Concerns

There are a two aspects of inclusion criteria that were not clear until fairly deep into the paper. First, the authors are only looking at adult outcomes. Second, included studies had to specifically distinguish effects for renters, not simply adjust for tenure or include an interaction. This would be good to mention early on.

The number of measures of housing insecurity is almost as high as the total number of included studies. Synthesizing the relationship between the constructs and mental health may be quite difficult with only 22 studies, since the exposures are so varied. If the effect varies across different exposures, it could be difficult to identify without more studies. Limiting to only particular exposures is not ideal, because that would cut the number of studies still. One possibility would be to broaden the criteria to include studies that do not distinguish renters. Another would be to examine other health outcomes beyond mental health. Another would be to include studies of children/adolescents. But this issue should be addressed either up front in the paper or in the discussion.

No timeframe is given for the literature search, although the earliest study considered was published in 2012. Should this be interpreted to mean that this topic was not studied prior to 2012, or was there an inclusion criterion that excluded earlier studies?

I was surprised to find that there were only 11 studies on this topic in the US. I think that may have to do with the inclusion criteria (i.e. this is only a small portion of the overall housing and health literature). In the literature search, was there an indication of which inclusion criterion was excluding most studies? The focus on mental health? The requirement that renters be separate? Adults only?

The authors assess strength of evidence, but they do not explicitly consider the causal nature of evidence presented in the included studies. It would be useful to have some sense of whether the comparison group used in the study represents a rigorous attempt to assess the causal effects of housing insecurity.

Is it useful to examine studies across so many countries, especially when the included countries have so few studies? It would be one thing if the search identified 10 studies from every country. But it is not clear that having one study from South Korea, one from Spain etc. is informative, particularly if the relationship between housing and health differs at all across countries. Since nearly all studies were from the US, UK, and Australia, the review could be limited to the anglosphere without losing much of the sample. At the very least, there should be a strong rationale for the inclusion of studies from all OECD countries.

6. PLOS authors have the option to publish the peer review history of their article (what does this mean? ). If published, this will include your full peer review and any attached files.

**Do you want your identity to be public for this peer review?** For information about this choice, including consent withdrawal, please see our Privacy Policy .

Reviewer #1: **Yes: ** Carme Borrell

Reviewer #2: No

---

## [Author Response · Author response to Decision Letter 1]

22 Mar 2024

Academic Editor: After careful consideration, we feel that it has merit but does not fully meet PLOS ONE’s publication criteria as it currently stands. Therefore, we invite you to submit a revised version of the manuscript that addresses the points raised during the review process. The reviewers have provided helpful information regarding an assortment of things concerning your manuscript. A consistent concern among both reviewers was the need to expand your lit review sources. Still, it appears that both reviewers feel the manuscript deserves some consideration once these edits are made.

Reply: Thank you for your guidance. We have carefully considered all suggestions and incorporated additional relevant literature to strengthen the manuscript. We are encouraged by the reviewers' acknowledgment of the manuscript's potential and refined it to meet PLOS ONE's publication criteria. We believe that these revisions have significantly improved the manuscript and hope it is now suitable for publication.

Reviewer #1: This article is very interesting. I make some comments on it:

Reply: Thank you for your insightful comments on our manuscript. We greatly appreciate your feedback and have made efforts to incorporate your suggestions diligently. Below, we outline the changes we have implemented in response to your comments.

Main comments

1. Introduction and discussion: To understand why access to housing is a problem nowadays it is important to present a conceptual framework of the causes, eg: public housing policies and the market. Many countries have left the access to housing to the market and therefore prices have increased and the accessibility has decreased. Moreover public spending on housing is very low. The paper should mention that. See for example:

-Marí-Dell'Olmo M, Novoa AM, Camprubí L, Peralta A, Vásquez-Vera H, Bosch J, Amat J, Díaz F, Palència L, Mehdipanah R, Rodríguez-Sanz M, Malmusi D, Borrell C. Housing Policies and Health Inequalities. Int J Health Serv. 2017 Apr;47(2):207-232. doi: 10.1177/0020731416684292. Epub 2016 Dec 28. PMID: 28030990.

Reply: Thank you for emphasizing the importance of presenting a conceptual framework to address the causes of housing access challenges. In response to your feedback, we have revised both the introduction and discussion (practical implications) to spotlight the primary macrostructural factors influencing housing insecurity and to offer an overview of the current state of housing accessibility in OECD countries.

Changes in manuscript (p. 3f.): “Housing insecurity is profoundly influenced by various macrostructural factors, including the welfare state, housing system, public housing policies, and market dynamics (Marí-Dell'Olmo et al., 2017). In their paper, Marí-Dell’Olmo et al. (2017) provide a conceptual framework on housing systems, housing conditions, and health equity, illustrating how these structural dimensions interact and impact access to adequate housing, as well as subsequent physical and mental health outcomes. Despite the diverse housing systems, regulatory environments, and socio-economic factors among countries, many advanced economies share a common trajectory of severe housing insecurity challenges (Lee et al., 2022). This review focuses on OECD countries for several reasons: They typically exhibit similar economic structures characterized by market-based economies with varying levels of government intervention, and they often adopt comparable housing policies and regulations, albeit with variations in their scope and implementation, such as social housing programs and tenant protections. Furthermore, in many OECD countries, heavy reliance on the market for access to housing has led to rising prices and reduced accessibility of housing, exacerbating the challenges for renters in securing affordable, stable, and suitable housing (Baumgartner et al., 2023; Bone, 2014; Clair et al., 2019; Eurostat, 2022; Routhier, 2019) (…) Moreover, public expenditure on housing remains low across many OECD countries, resulting in a heavy reliance on the often minimally regulated private rental market for access to affordable housing (Del Salvi Peroi et al., 2016). Although some governments have implemented a range of housing policies and programs to address the issue of housing security, including social/public housing, housing subsidies, or eviction policies, these initiatives frequently fail to reach those who truly need them, leaving many eligible renters without the necessary support (DeLuca & Rosen, 2022; Dweik & Woodhall-Melnik, 2022) (…) within OECD countries.“

Changes in manuscript (p. 34f.): “At the same time, our findings emphasize the critical need to enhance access to affordable and stable housing options for renters, especially low-income households, as a means to combat housing insecurity and enhance public mental health. Structural factors such as welfare state policies, housing systems, and market dynamics significantly influence housing insecurity and insufficient public expenditure on housing exacerbates this issue (Del Salvi Peroi et al., 2016; Lee et al., 2022; Marí-Dell'Olmo et al., 2017). Therefore, prioritizing policy responses to address housing insecurity is vital for promoting the well-being and resilience of affected individuals and communities. However, identifying effective housing policy measures, particularly for vulnerable renter groups such as low-income, disabled, or racially marginalized households, requires further research and investigation (DeLuca & Rosen, 2022).”

2. Introduction, results and discussion: Housing insecurity change by the axes of inequality (gender, social class, race…). It is necessary to deeper emphasize this reality. Do the papers found analyse this fact? See for example:

-Vásquez-Vera C, Fernández A, Borrell C. Gender-based inequalities in the effects of housing on health: A critical review. SSM Popul Health. 2022 Mar 11;17:101068. doi: 10.1016/j.ssmph.2022.101068. PMID: 35360438; PMCID: PMC8961216.

Reply: Thank you for your highlighting the fact that inequalities such as gender, social class, and race intersect with housing insecurity and its impact on health outcomes. We have underscored this intersection in our introduction and provided a more detailed discussion outline: Regrettably, the studies included in our analysis mostly did not specifically address or explore this aspect but rather examined the entire group of renters collectively. We acknowledge this as a significant gap in the current literature and have emphasized it in our text to suggest avenues for future research in response to your commentary in more detail (both in the introduction and discussion).

Changes in manuscript (p. 4): “(…) as well as dimensions of inequality and discrimination such as social class, race, disability, and gender, creating a complex interplay in which these interact and mutually reinforce each other (Hulse & Saugeres, 2008; C. Vásquez-Vera et al., 2022).”

Changes in manuscript (p. 33): “(…) such as gender, social class, disability, and race (Bentley et al., 2019; Carrere et al., 2022; DeLuca & Rosen, 2022; Hulse & Saugeres, 2008; Kirkpatrick & Tarasuk, 2011; Morris et al., 2017; C. Vásquez-Vera, Fernández, & Borrell, 2022). (…) For instance, in a review examining gender-based inequalities in the effects of housing on health, findings suggest that while negative impacts are observed across all genders regarding housing affordability, instability, and insecurity, women, non-binary, and transgender people tend to experience more pronounced effects (C. Vásquez-Vera, Fernández, & Borrell, 2022). However, the studies included in our analysis exclusively examined renters as a whole. Recognizing the heterogeneity within this group and the likelihood of varied experiences of housing insecurity and its impact on mental health based on economic and social circumstances, it is essential for future research to adopt an intersectional lens and a bidirectional approach (cf. conceptual framework proposed by C. Vásquez-Vera, Fernández, and Borrell, 2022). Studies further indicate that housing insecurity and its impact on mental health also varies on a macro-level, across different social, legal, and cultural contexts (Acolin, 2022; Bentley et al., 2016; Herbers & Mulder, 2017; Hulse & Milligan, 2014; Hulse et al., 2011; C. Vásquez-Vera, Fernández, & Borrell, 2022) (…).”

3. Results: I am not sure that tables 1 and 2 have to be inside the paper, they can be in an appendix. In the results section I would include a table showing the descriptive results found: the number of papers for each variable (including the mental health results).

Reply: We agree that the results section benefits from a descriptive table, as tables 1 and 2 are quite long and may make it challenging for the reader to quickly grasp the descriptive results. Therefore, we have followed your proposal and added a table showing the descriptive results. Regarding the placement of tables 1 and 2 (now tables 2 and 3), we defer to the editor's decision. However, personally, we would opt to leave these tables inside the paper, as advised in the PRISMA guidelines, as they provide essential information for our paper.

Changes in manuscript (p. 11): see Table 1

4. Discussion: an important limitation of the paper is the language chosen by the review. They exist articles in other languages that are lost. The article should mention this limitation.

Reply: Thank you for pointing out this important limitation. We have added it to our limitations.

Changes in manuscript (p. 35): “Fifth, this review is limited to publications in English, potentially overlooking valuable studies in other languages. Although we excluded only two studies based on language, given that our keywords were restricted to English, it is possible that the actual number of papers on this topic is higher.”

Other minor comments:

5. Methods: please explain the meaning of “data points”. Does it mean cross sectional data?

Reply: Thank you for pointing out this misleading formulation. We meant by it that studies may include analyses of other populations (e.g., homeowners, children) and research questions (e.g., other housing dimensions) if they provided separate results or analysis for renters. In this case, only data relevant to our research focus on renters was extracted for the review. We specified this in the method section.

Changes in manuscript (p. 6f.): „studies“ instead of „data points“

Changes in manuscript (p. 7): “Studies may include analyses of other populations (e.g., homeowners, children) and research questions (e.g., other housing dimensions), if they provided separate analysis for our research focus. However, studies needed to specifically distinguish effects for renters, not simply adjust for tenure or include an interaction.”

6. Explain the meaning of “PTSD”.

Reply: Thank you for pointing out the missing definition, which we added in the manuscript.

Changes in manuscript (p. 11, 21, 25, 29): “…post-traumatic stress disorder…”

Carme Borrell

Reply: We hope these revisions adequately address your concerns and enhance the clarity and comprehensiveness of our manuscript. We are grateful for your valuable feedback!

Reviewer #2: This is an important topic with a growing literature, and it is natural to have a review piece that synthesizes the increasing evidence for the relationship between housing insecurity and mental health. The authors identify 22 studies examining the relationship between some measure of housing affordability or stability and mental health. There are several strengths of the review. First, the authors consider multiple dimensions of housing insecurity. The authors also make some attempt to assess the evidence quality or describe the study methodology. Although I think the motivation for a review piece on this topic is clear, I have some concerns about the approach used, and am somewhat skeptical the authors have identified an exhaustive list of studies.

Reply: Thank you for stressing the importance of the topic and the relevance of a systematic review on the relationship between housing insecurity and mental health. We are grateful for the thorough review of the manuscript and appreciate your suggestions and concerns. We have changed the manuscript accordingly with a focus on aiming to strengthen the line of arguments, giving more details about our approach, and – where necessary adding clarifications and limitations. Below, we outline the changes we have implemented in response to your comments in detail:

Main Concerns

1. There are a two aspects of inclusion criteria that were not clear until fairly deep into the paper. First, the authors are only looking at adult outcomes. Second, included studies had to specifically distinguish effects for renters, not simply adjust for tenure or include an interaction. This would be good to mention early on.

Reply: Thank you for your feedback regarding the lack of clarity of our inclusion criteria and the importance of explicitly stating our focus on adult outcomes as well as the requirement for studies to distinguish effects for renters early on. We have adapted the manuscript to clarify these aspects earlier in the paper to enhance transparency and understanding for readers. With regard to the fact that we only look at adult outcomes, this target population is additionally mentioned in the abstract (p. 1), objectives (p. 6), and methods (p. 7).

Changes in manuscript (p. 1, abstract): “Studies needed to specifically distinguish effects for renters, not simply adjust for tenure or include an interaction.”

Changes in manuscript (p. 3 + p. 4, introduction): “… adult ...”

Changes in manuscript (p. 7, methods): “Studies may include analyses of other populations (e.g., homeowners, children) and research questions (e.g., other housing dimensions), if they provided separate analysis for our research focus. However, studies needed to specifically distinguish effects for renters, not simply adjust for tenure or include an interaction.”

2. The number of measures of housing insecurity is almost as high as the total number of included studies. Synthesizing the relationship between the constructs and mental health may be quite difficult with only 22 studies, since the exposures are so varied. If the effect varies across different exposures, it could be difficult to identify without more studies. Limiting to only particular exposures is not ideal, because that would cut the number of studies still. One possibility would be to broaden the criteria to include studies that do not distinguish renters. Another would be to examine other health outcomes beyond mental health. Another would be to include studies of children/adolescents. But this issue should be addressed either up front in the paper or in the discussion.

Reply: Thank you for the important suggestions. We have addressed the number of measures and exposures and the possibilities to broaden the criteria, examine additional health outcomes, or going beyond adult samples, by adding the paragraph below to our discussion section. In addition, we refer to the housing literature to highlight the potential benefit of adopting a unified definition of housing instability and corresponding measures in future studies (p. 32).

Changes in manuscript (p. 35): “Specifically, the operationalizations of housing affordability and instability, as well as mental health outcomes, varied significantly across the included studies (see Tables 1 and 4). However, due to the limited number of studies, discerning which housing insecurity factors and/or mental health outcomes are more or less relevant remains challenging. Confining the analysis to particular variables and operationalizations would have significantly reduced the pool of studies available for analysis. A potential avenue for future research could involve broadening the inclusion criteria to encompass studies that do not differentiate tenure or expanding the range of included studies by including physical health outcomes.”

3. No timeframe is given for the literature search, although the earliest study considered was published in 2012. Should this be interpreted to mean that this topic was not studied prior to 2012, or was there an inclusion criterion that excluded ear

---

## [Decision Letter · Decision Letter 1]

5 Aug 2024

PONE-D-23-35202R1Exploring the association between housing insecurity and mental health among renters: A systematic reviewPLOS ONE

Dear Dr. Talmatzky,

Thank you for submitting your manuscript to PLOS ONE. After careful consideration, we feel that it has merit but does not fully meet PLOS ONE’s publication criteria as it currently stands. Therefore, we invite you to submit a revised version of the manuscript that addresses the points raised during the review process.

During internal editorial checks on this submission, a previously unaddressed concern was noted. Specifically, we note that you have used reporting checklists such as STROBE to assess the quality of included articles. Conclusions in a systematic review or meta-analysis should be related to the quality of the included publications, but this assessment should relate to the conduct of the study, not the reporting. Therefore, before we can consider publishing your article, we require that you conduct a more appropriate quality assessment, and frame your conclusions based on the results of such an assessment. A potentially useful repository of quality assessment tools for different study types is available at https://osf.io/dmrq6

In addition, we note that your systematic review was restricted to publications in English. Given that the inclusion criteria includes all OECD countries, and there is a possibility that research relating to local policy may well be published in national journals, we recommend that you remove this restriction, or else provide a strong scientific justification for not doing so.

Please also provide details of the level of agreement between reviewers, given that the screening and extraction were not completely conducted in duplicate. If this was <80%, we suggest that all screening and extraction be duplicated to ensure agreement.

We look forward to receiving your revised manuscript.

Kind regards,

Marianne Clemence, Staff Editor, on behalf of,

Andrew T. Carswell

Academic Editor

PLOS ONE

Additional Editor Comments:

Comments from PLOS Editorial Office: We note that one or more reviewers has recommended that you cite specific previously published works in an earlier round of revision. As always, we recommend that you please review and evaluate the requested works to determine whether they are relevant and should be cited. It is not a requirement to cite these works and you may remove them before the manuscript proceeds to publication. We appreciate your attention to this request.

Reviewers' comments:

Reviewer's Responses to Questions

**Comments to the Author**

1. If the authors have adequately addressed your comments raised in a previous round of review and you feel that this manuscript is now acceptable for publication, you may indicate that here to bypass the “Comments to the Author” section, enter your conflict of interest statement in the “Confidential to Editor” section, and submit your "Accept" recommendation.

Reviewer #2: All comments have been addressed

2. Is the manuscript technically sound, and do the data support the conclusions?

Reviewer #2: Yes

3. Has the statistical analysis been performed appropriately and rigorously? 

Reviewer #2: Yes

4. Have the authors made all data underlying the findings in their manuscript fully available?

Reviewer #2: Yes

5. Is the manuscript presented in an intelligible fashion and written in standard English?

Reviewer #2: (No Response)

6. Review Comments to the Author

Reviewer #2: The authors have responded well to my concerns and the manuscript is improved. I still think it is alarming that there are only 22 studies published ever, and that the majority were published since 2020. Maybe worth explicitly mentioning as a finding or implication, but not totally necessary.

7. PLOS authors have the option to publish the peer review history of their article (what does this mean? ). If published, this will include your full peer review and any attached files.

**Do you want your identity to be public for this peer review?** For information about this choice, including consent withdrawal, please see our Privacy Policy .

Reviewer #2: No

---

## [Author Response · Author response to Decision Letter 2]

30 Oct 2024

Dear Dr. Carswell,

Dear Marianne Clemence,

We appreciate your additional feedback and the opportunity to submit a revised version of our manuscript titled "Exploring the association between housing insecurity and mental health among renters: A systematic review." We addressed your suggestions in a careful process of revision and will describe all changes in detail in this response letter. Additionally, we utilized the track changes mode of MS Office for all content modifications.

A major change from previous versions of the manuscript was a change in authorship. Co-author Laura Nohr conducted the quality assessment of all included studies and made most of the changes in the manuscript. Mira Talmatzky and Helen Niemeyer reviewed carefully the changes made and performed the second quality rating of 50% of the included studies. Based on this and her previous work on earlier versions of the manuscript, we decided to share the first authorship between Mira Talmatzky and Laura Nohr. We have also highlighted this change in the manuscript.

We hope that we could respond to all comments and recommendations to your satisfaction. We believe that these revisions have significantly strengthened the manuscript and hope it is now suitable for publication.

If there are any remaining questions or concerns, please do not hesitate to contact us.

Best regards,

Mira Talmatzky & Laura Nohr

(on behalf of all authors)

Editor: During internal editorial checks on this submission, a previously unaddressed concern was noted. Specifically, we note that you have used reporting checklists such as STROBE to assess the quality of included articles. Conclusions in a systematic review or meta-analysis should be related to the quality of the included publications, but this assessment should relate to the conduct of the study, not the reporting. Therefore, before we can consider publishing your article, we require that you conduct a more appropriate quality assessment, and frame your conclusions based on the results of such an assessment. A potentially useful repository of quality assessment tools for different study types is available at https://osf.io/dmrq6

Response: Thank you for your careful review of our manuscript and study conduction. We understand the additional benefits of quality assessments compared to reporting checklists. Therefore, we decided to apply the JBI critical appraisal tools for analytical cross-sectional and cohort studies and rated 50% twice. Overall, the methodological quality of included studies was good. In the results section we describe strengths and weaknesses of studies and potential risks of bias. In appendix S4 we provide details of the assessment.

Editor: In addition, we note that your systematic review was restricted to publications in English. Given that the inclusion criteria includes all OECD countries, and there is a possibility that research relating to local policy may well be published in national journals, we recommend that you remove this restriction, or else provide a strong scientific justification for not doing so.

Response: We appreciate your insightful comment and followed your recommendation. After removing the restriction to publications in English, two additional studies became eligible for inclusion – one study in German and one study in Spanish. In the end, both were not included in our review because of (a) a lack of housing insecurity or housing affordability measures and (b) not studying exclusively renters. We documented the inclusion and exclusion carefully and updated the respective appendix.

Editor: Please also provide details of the level of agreement between reviewers, given that the screening and extraction were not completely conducted in duplicate. If this was <80%, we suggest that all screening and extraction be duplicated to ensure agreement.

Response: Thank you for pointing out the importance of reporting the level of agreement. The interrater reliability was already reported on page 10. Now we are also reporting the interrater agreement on data extraction (92.1%) and quality assessment (89.0%). We would argue that we reached satisfying level of agreement in both cases.

We hope we have addressed your concerns satisfactorily. Thank you again for your valuable feedback! Your contributions have helped us clarify important aspects and strengthen the overall presentation of our research.

---

## [Decision Letter · Decision Letter 2]

21 Jan 2025

PONE-D-23-35202R2Exploring the association between housing insecurity and mental health among renters: A systematic reviewPLOS ONE

Dear Dr. Talmatzky,

Thank you for submitting your manuscript to PLOS ONE. After careful consideration, we feel that it has merit but does not fully meet PLOS ONE’s publication criteria as it currently stands. Therefore, we invite you to submit a revised version of the manuscript that addresses the points raised during the review process.

Thank you for revising your manuscript to address editorial concerns about the conduct of the systematic review. Given the potential for general interest in these findings, and to ensure a rigorous review, the revised version has now been assessed by a member of our statistical advisory group. The reviewer has provided a number of further recommendations to improve the quality of the manuscript and ensure it adheres to community standards for systematic reviews. Updating the searches at this time can be considered optional, although it would likely increase the relevance of the work, but the other concerns should be addressed since they relate to our publication criteria.

I appreciate that additional revisions at this stage are frustrating, but I hope you understand the reasons for this decision.

We look forward to receiving your revised manuscript.

Kind regards,

Marianne Clemence, Staff Editor, on behalf of,

Andrew T. Carswell

Academic Editor

PLOS ONE

Reviewers' comments:

Reviewer's Responses to Questions

**Comments to the Author**

1. If the authors have adequately addressed your comments raised in a previous round of review and you feel that this manuscript is now acceptable for publication, you may indicate that here to bypass the “Comments to the Author” section, enter your conflict of interest statement in the “Confidential to Editor” section, and submit your "Accept" recommendation.

Reviewer #3: (No Response)

2. Is the manuscript technically sound, and do the data support the conclusions?

Reviewer #3: Partly

3. Has the statistical analysis been performed appropriately and rigorously? 

Reviewer #3: N/A

4. Have the authors made all data underlying the findings in their manuscript fully available?

Reviewer #3: Yes

5. Is the manuscript presented in an intelligible fashion and written in standard English?

Reviewer #3: Yes

6. Review Comments to the Author

Reviewer #3: GENERAL COMMENTS

Thank you for the opportunity to review this revised systematic review titled “Exploring the association between housing insecurity and mental health among renters: A systematic review”. I note here that I have not reviewed this manuscript before and did not look at the prior reviewer’s comments so as to not bias my own review. While I think this is a critically important topic and is generally well done, I have several suggestions that I believe will strengthen this work, detailed in my specific comments that follow. The two major issues that I see are the need to update the search for potentially eligible studies since it is almost two years old now, as well as the need to use a formal instrument to assess the strength/certainty of evidence. I focus my review primarily, but not exclusively, on the systematic review methodology, something that inextricably affects the results and discussion sections of any type of study.

I am hopeful that my comments will be taken in the spirit of improving this work, with the specific goal of advancing the field. Also, please note that my comments are focused on the clean copy of the revised manuscript.

SPECIFIC COMMENTS

*Title page – Per PRISMA 2020 guidelines, please include the type(s) of study design(s) that were eligible in your title.

*Page 1, lines 24 through 49 (Abstract) – First, while working within the guidelines of the journal, please revise your abstract in accordance with PRISMA 2020 guidelines, void of the meta-analytic piece (See, for example: Page MJ, Moher D, Bossuyt PM, et al. PRISMA 2020 explanation and elaboration: updated guidance and exemplars for reporting systematic reviews. BMJ. 2021;372:n160. https://www.bmj.com/content/bmj/372/bmj.n160.full.pdf)

The 12-item checklist (Box 2) on the third page of the PRISMA guidelines should be helpful to you. Second, on line 25, suggest that you delete the word “first” or say something like “To the best of the authors knowledge, this is the first…” My main point here is that using the term “first” is usually too verbose given the margin of search error when trying to identify related systematic reviews on a topic. Third, on lines 28 and 29 I see that the searches were conducted in December of 2022. Since the search for studies is approximately two years old, and despite the fact that you performed forward citation tracking up to February 27, 2023, the search for potentially eligible studies probably needs to be updated. Fourth, and per PRISMA guidelines, you should briefly report the results of your risk of bias assessments. You should probably also assess the certainty of evidence using something like the Grading of Recommendations Assessment, Development, and Evaluation (GRADE) instrument. Guidelines for doing so when a meta-analysis is not conducted are available here: Murad MH, Mustafa RA, Schünemann HJ, Sultan S, Santesso N. Rating the certainty in evidence in the absence of a single estimate of effect. Evidence Based Medicine. 2017;22(3):85-87. https://ebm.bmj.com/content/ebmed/22/3/85.full.pdf. Finally, your conclusions should be tempered by the certainty of evidence and most likely the need for future well-designed studies on this topic.

*Pages 2 through 5, lines 51 through 150 (Introduction) – Overall, a nice Introduction section.

*Page 6, lines 163 through 167 (Methods) – Please tell the reader if the decision to conduct a systematic review without a meta-analysis was made a priori or post hoc. Along those lines, throughout the rest of your Methods section, please tell the reader what post hoc changes were made to your a priori protocol as well as the reasons for such. This is of course assuming that you had an a priori protocol.

*Pages 6 and 7, lines 169 through 193 (Methods, Eligibility criteria) – First, please make it clearer to the reader if eligible studies were limited to those published in peer-reviewed journals or studies from other sources (masters theses, dissertations, abstracts from conference proceedings, etc.) were also eligible. Second, in the first paragraph, you use the term “such as” in multiple places. However, your eligibility criteria should be more definitive.

*Pages 7 and 8, lines 195 through 216 (Methods, Search Strategy) – Please tell the reader who was responsible for conducting the database searches. Also, and as previously mentioned, the search for studies should be updated.

*Page 8, lines 218 through 231 (Methods, Study Selection) – First, on line 219, please tell the reader if duplicates were removed electronically and/or manually. Second, on lines 230-231, the use of Gwet’s AC1 statistic may be preferable to Cohen’s kappa. See, for example: 1. Gwet KL. Computing inter-rater reliability and its variance in the presence of high agreement. Br J Math Stat Psychol. 2008;61(Pt 1):29-48. https://doi.org/10.1348/000711006x126600

*Pages 8 and 9, lines 233 through 255 (Methods, Data Extraction and Synthesis Method) - Please tell the reader if any attempt was made to retrieve missing data from investigator. If not, then this should also be stated. If missing data were requested, the process for doing so should be described. Also, on line 246, interrater agreement may also be better assessed using Gwet’s AC1 statistic.

*Pages 9 and 10, lines 257 through 263 (Methods, Reporting Quality) – First, suggest that you retitle this subsection as “Risk of bias assessment”. Second, why did you not use something like the MASTER scale, an assessment instrument that allows for the evaluation of all the different types of study designs you included versus two separate instruments? See: 1. Stone JC, Glass K, Clark J, et al. The MethodologicAl STandards for Epidemiological Research (MASTER) scale demonstrated a unified framework for bias assessment. J Clin Epidemiol. 2021;134:52-64. Doi: 10.1016/j.jclinepi.2021.01.012; 2. Ahmed AI, Kaleem MZ, Elshoeibi AM, et al. MASTER scale for methodological quality assessment: Reliability assessment and update. 2024. J Evid Based Med. doi: 10.1111/jebm.12618. Third, on line 263 interrater agreement may also be better assessed using Gwet’s AC1 statistic.

After line 263, two other titled subsections should probably be added. The first should be titled something like “Strength/Certainty of Evidence” and assessed using something like the previously mentioned Grading of Recommendations Assessment, Development, and Evaluation (GRADE) instrument. Guidelines for doing so when a meta-analysis is not conducted are available here: Murad MH, Mustafa RA, Schünemann HJ, Sultan S, Santesso N. Rating the certainty in evidence in the absence of a single estimate of effect. Evidence Based Medicine. 2017;22(3):85-87. https://ebm.bmj.com/content/ebmed/22/3/85.full.pdf. Please note that this section is different than the assessment for risk of bias. In fact, risk of bias assessments feed into the GRADE assessment. Also, more than one person should be responsible for assessing the strength/certainty of evidence.

The second subsection that should be added could be titled something like “Research Synthesis”. This is where you describe the fact that you analyzed your results qualitatively, i.e., without a meta-analysis, as well as telling the reader why you did not conduct a meta-analysis. You should also tell the reader that you limited your statistics to those that were descriptive in nature, i.e., frequencies and percentages. Some of this information was previously described earlier in your Methods but should be moved to this final subsection of your Methods.

*Page 10, line 264 – Strongly suggest that you replace the title “Evidence Synthesis” with “Results”.

*Page 10, line 276 (Study Selection) – Here and throughout the rest of the manuscript, please reference the 22 studies that were included, whether discussed in part or in full.

*Table 1, pages 11 and 12 – This is a nice table.

*Table 2, pages 13 through 16 – This is a nice table. However, I would suggest that you also add a column that includes any funding that these studies received.

*Table 3, pages 17 through 23 – This is another nice table. My only suggestion is that for each study you delineate exactly what variables the original study authors considered to be primary as well as what variables were considered secondary.

*Table 4, pages 24 and 25 – This is yet another nice table.

*Pages 24 through 31, lines 309 through 477 (Results) – With the exception of the need for better referencing, this section of results are well-described

*Pages 31 through 37, lines 479 through 640 (Discussion) – For ease of reading, suggest that you partition your Discussion into the following subsections and discuss: (1) Overall findings, (2) Implications for Research, (3) Implications for Practice, (4) Implications for Policy (if any), and (5) Strengths and Potential Limitations. Also, your discussion regarding the strength of evidence on pages 33 through 34 (lines 524 through 567) would be strengthened by formal assessment using something like the GRADE instrument I recommended earlier in my review.

*Page 37, lines 642 through 650 (Conclusions) – First, and as previously mentioned, I would delete the word “first” on line 642 or revise to say something like “To the best of the authors knowledge, this is the first systematic review…” Second, and as also previously mentioned, I think your Conclusions need to be tempered after formal assessment of the strength/certainty of evidence and most likely the need for future well-designed studies on this topic.

*Pages 38 through 44 (References) – Suggest that you place your references in the same font and size as your manuscript, adhere to the journal guidelines, and be consistent. For example, some journal titles are spelled out in full while others are abbreviated.

*S1 (PRISMA Checklist) – Some items in the manuscript and other materials are either not reported or not reported in sufficient detail.

END OF REVIEW

7. PLOS authors have the option to publish the peer review history of their article (what does this mean? ). If published, this will include your full peer review and any attached files.

**Do you want your identity to be public for this peer review?** For information about this choice, including consent withdrawal, please see our Privacy Policy .

Reviewer #3: No

---

## [Author Response · Author response to Decision Letter 3]

2 Jun 2025

PONE-D-23-35202_R3

Exploring the association between housing insecurity and mental health among renters: A systematic review

Dear Dr. Clemence,

Dear Dr. Carswell,

Dear Reviewer,

Thank you for your feedback on our manuscript and the additional communication. We would like to reply to the reviewer and to the additional communication in this rebuttal letter.

We still think that the whole review process has not been transparent and comprehensible for us as authors. Probably, we would have preferred to receive the expert’s opinion during the first additional revision after the formal acceptance of the manuscript. However, since we have already put a lot of efforts into the manuscript and think that it targets an important topic, we would like to submit a revised version of the manuscript. As suggested, we decided not to update the literature search since we think that we submitted the review timely after the search.

I am very sorry that the submission of the revised manuscript has been delayed on my part. A long period of sick leave did not allow me to complete the revision earlier.

Now, we would like to respond to the reviewer’s comments:

PLOS ONE

Reviewers' comments:

Reviewer's Responses to Questions

Comments to the Author

1. If the authors have adequately addressed your comments raised in a previous round of review and you feel that this manuscript is now acceptable for publication, you may indicate that here to bypass the “Comments to the Author” section, enter your conflict of interest statement in the “Confidential to Editor” section, and submit your "Accept" recommendation.

Reviewer #3: (No Response)

2. Is the manuscript technically sound, and do the data support the conclusions? The manuscript must describe a technically sound piece of scientific research with data that supports the conclusions. Experiments must have been conducted rigorously, with appropriate controls, replication, and sample sizes. The conclusions must be drawn appropriately based on the data presented.

Reviewer #3: Partly

3. Has the statistical analysis been performed appropriately and rigorously?

Reviewer #3: N/A

4. Have the authors made all data underlying the findings in their manuscript fully available?

The PLOS Data policy requires authors to make all data underlying the findings described in their manuscript fully available without restriction, with rare exception (please refer to the Data Availability Statement in the manuscript PDF file). The data should be provided as part of the manuscript or its supporting information, or deposited to a public repository. For example, in addition to summary statistics, the data points behind means, medians and variance measures should be available. If there are restrictions on publicly sharing data—e.g. participant privacy or use of data from a third party—those

must be specified.

Reviewer #3: Yes

5. Is the manuscript presented in an intelligible fashion and written in standard English? PLOS ONE does not copyedit accepted manuscripts, so the language in submitted articles must be clear, correct, and

unambiguous. Any typographical or grammatical errors should be corrected at revision, so please note any specific errors here.

Reviewer #3: Yes

6. Review Comments to the Author

Reviewer #3: GENERAL COMMENTS

Thank you for the opportunity to review this revised systematic review titled “Exploring the association between housing insecurity and mental health among renters: A systematic review”. I note here that I have not reviewed this manuscript before and did not look at the prior reviewer’s comments so as to not bias my own review. While I think this is a critically important topic and is generally well done, I have several suggestions that I believe will strengthen this work, detailed in my specific comments that follow. The two major issues that I see are the need to update the search for potentially eligible studies since it is almost two years old now, as well as the need to use a formal instrument to assess the strength/ certainty of evidence. I focus my review primarily, but not exclusively, on the systematic review methodology, something that inextricably affects the results and discussion sections of any type of study.

I am hopeful that my comments will be taken in the spirit of improving this work, with the specific goal of advancing the field. Also, please note that my comments are focused on the clean copy of the revised manuscript.

Thank you very much for taking the time to review our manuscript. The constructive suggestions have helped us to improve the manuscript considerably. And yes, we took all of your feedback as a chance to improve our work and to contribute to the field of study. Further, we appreciate your concerns to review only the clean version of the manuscript and not read the previous reviews. However, we think it would be helpful to give some context to the review process so far. Our manuscript had been formally accepted in July 2024 and the Editor decided afterwards to retract the acceptance. Since then, we have put a lot of efforts into the improvement of the manuscript. This is now the third round of revision, each with different requirements and partly contradicting suggestions to previous reviews. With this history of revision process in mind, we will answer to your suggestions.

SPECIFIC COMMENTS

*Title page – Per PRISMA 2020 guidelines, please include the type(s) of study design(s) that were eligible in your title.

Thank you for your feedback, we changed the title accordingly:

Exploring the association between housing insecurity and mental health among renters: A systematic review of quantitative primary and secondary studies (Titlepage, l. 8-10)

*Page 1, lines 24 through 49 (Abstract) – First, while working within the guidelines of the journal, please revise your abstract in accordance with PRISMA 2020 guidelines, void of the meta-analytic piece (See, for example: Page MJ, Moher D, Bossuyt PM, et al. PRISMA 2020 explanation and elaboration: updated guidance and exemplars for reporting systematic reviews. BMJ. 2021;372:n160. https://www.bmj.com/content/bmj/372/bmj.n160.full.pdf).The 12-item checklist (Box 2) on the third page of the PRISMA guidelines should be helpful to you.

Thank you very much for your detailed feedback and guidance. We changed the abstract accordingly and indicated which information refers to which of the 12 items of the checklist (p.1 – 2, l. 24 – 62).

Second, on line 25, suggest that you delete the word “first” or say something like “To the best of the authors knowledge, this is the first…” My main point here is that using the term “first” is usually too verbose given the margin of search error when trying to identify related systematic reviews on a topic.

Thank you for this important hint. We changed the sentence as suggested:

“To the best of the authors’ knowledge, the present systematic review is the first to investigate the association between housing insecurity and mental health outcomes among renters, with a focus on housing affordability and instability.” (p. 1, l. 25 – 28)

Third, on lines 28 and 29 I see that the searches were conducted in December of 2022. Since the search for studies is approximately two years old, and despite the fact that you performed forward citation tracking up to February 27, 2023, the search for potentially eligible studies probably needs to be updated.

Thank you for the feedback on the date of our searches. We agree that the searches could benefit from an update. Due to the long review process and the different obstacles, we have been facing during the process, we decided to accept the Editor’s offer not to update the searches.

Fourth, and per PRISMA guidelines, you should briefly report the results of your risk of bias assessments. You should probably also assess the certainty of evidence using something like the Grading of Recommendations Assessment, Development, and Evaluation (GRADE) instrument. Guidelines for doing so when a meta-analysis is not conducted are available here: Murad MH, Mustafa RA, Schünemann HJ, Sultan S, Santesso N. Rating the certainty in evidence in the absence of a single estimate of effect. Evidence Based Medicine. 2017;22(3):85-87. https://ebm.bmj.com/content/ebmed/22/3/85.full.pdf.

Thank you for your recommendation. We applied the GRADE instrument to assess certainty of evidence for each mental health outcome according to the procedure recommended by Murad et al. (2017). All outcomes were rated independently by two raters. We describe the procedure in the method section (p. 11, l. 293 – 306) and present the results in the results section (p. 31 – 34, l. 471 – 527).

Further, we added the respective information to the Abstract:

“The methodological quality of the included studies was rated with the JBI Critical Appraisal Tools, and the certainty of evidence was rated using the Grading of Recommendations Assessment, Development, and Evaluation (GRADE) framework.” (p. 2, l. 38 – 41)

“Based on the JBI ratings, the overall methodological quality of the included studies was good. The overall ratings of certainty of evidence, based on the GRADE ratings, were between low and very low – mainly due to the non-controlled study designs of included studies.” (p. 2, l. 45 – 47)

Finally, your conclusions should be tempered by the certainty of evidence and most likely the need for future well-designed studies on this topic.

Thank you for this important suggestion. We used the results of the GRADE rating to discuss the findings and our conclusions accordingly (compare p. 37 – 47). Based on a mostly low certainty of evidence, we now emphasize the need for future well-designed studies. We revised the abstract accordingly:

“Despite methodological limitations due to the non-controlled studies included in the review, the findings suggest overall that experiencing unaffordable or unstable housing has a negative impact on renters’ overall mental health and depressive symptoms. Housing insecurity poses a significant challenge for renters in OECD countries, highlighting the need for policymakers to implement supportive housing policies and tenure protection measures in order to improve renters’ housing security and ultimately public health. Nevertheless, more research with robust study designs is needed to draw further conclusions.” (p. 2, l. 54 – 61)

*Pages 2 through 5, lines 51 through 150 (Introduction) – Overall, a nice Introduction section.

Thank you for your positive feedback on our Introduction section.

*Page 6, lines 163 through 167 (Methods) – Please tell the reader if the decision to conduct a systematic review without a meta-analysis was made a priori or post hoc. Along those lines, throughout the rest of your Methods section, please tell the reader what post hoc changes were made to your a priori protocol as well as the reasons for such. This is of course assuming that you had an a priori protocol.

Although we did not publish a study protocol or a preregistration, we followed a strict a priori defined procedure during the systematic review. It was determined a priori that the decision on whether to conduct a meta-analysis would be made post hoc after reviewing the suitability of the data:

“Based on our knowledge of the field of research and anticipated heterogeneity in study methodologies and definitions of housing insecurity, we decided a priori that a meta-analysis would be considered if sufficient homogeneity was found—which was ultimately not the case. Therefore, a narrative synthesis was chosen over a meta-analysis to synthesize the evidence.” (p. 12, l. 311 – 314)

*Pages 6 and 7, lines 169 through 193 (Methods, Eligibility criteria) – First, please make it clearer to the reader if eligible studies were limited to those published in peer-reviewed journals or studies from other sources (masters theses, dissertations, abstracts from conference proceedings, etc.) were also eligible.

Thank you for pointing this out. We added more specific information regarding the publication status of eligible studies to the manuscript:

“No restrictions were made regarding publication date or status including published and unpublished studies, partially published studies, and studies from other sources like masters’ theses, dissertations, abstracts from conference proceedings, not peer reviewed.” (p. 8, l. 204 – 207)

Second, in the first paragraph, you use the term “such as” in multiple places. However, your eligibility criteria should be more definitive.

Thank you for this suggestion. We aimed at giving examples to illustrate the respective inclusion criteria. We changed the wording accordingly and added the relevant operationalization of the respective construct in parenthesis:

“To be eligible for inclusion, studies needed to be quantitative primary or secondary research that investigated housing insecurity by examining at least one variable related to housing affordability (housing cost burden, housing affordability stress, receiving rental assistance, or living in social housing), and/or housing instability (frequent moves, forced moves, risk of eviction, behind on rent, or overcrowding). Studies investigating housing programs or policies aimed at improving affordability and/or stability (rental assistance or eviction prevention programs), were also considered for inclusion. Furthermore, studies needed to include at least one mental health-related outcome (specific mental disorders, symptoms of psychological distress, overall mental health and well-being, quality of life, psychosocial functioning, suicidal behavior, or treatment for mental health reasons, e.g., medication, outpatient, or inpatient therapy for emotional conditions).” (p. 7 – 8, l. 190 – 200)

*Pages 7 and 8, lines 195 through 216 (Methods, Search Strategy) – Please tell the reader who was responsible for conducting the database searches. Also, and as previously mentioned, the search for studies should be updated.

Thank you for your recommendations. As described above, we decided not to update the search for studies. However, we stated explicitly the responsibility for conducting the database searches:

“A comprehensive search of the literature was conducted in two stages by the first author M.T. under the supervision of the last author H.N.” (p. 8, l. 220 – 221)

*Page 8, lines 218 through 231 (Methods, Study Selection) – First, on line 219, please tell the reader if duplicates were removed electronically and/or manually.

Thank you for this recommendation. We added the information as suggested:

“After exporting all search results into the reference management program Citavi (Version 6.14), duplicate records were removed manually by the first author M.T.” (p. 9, l. 244 – 245)

“The initial search yielded a total of 1,198 potentially eligible titles (after the first author M.T. had removed duplicates manually).” (p. 12, l. 321 – 323)

Second, on lines 230-231, the use of Gwet’s AC1 statistic may be preferable to Cohen’s kappa. See, for example: 1. Gwet KL. Computing inter-rater reliability and its variance in the presence of high agreement. Br J Math Stat Psychol. 2008;61(Pt 1):29-48.

Thank you for this helpful suggestion. We followed your advice and calculated Gwet’s AC1 instead of Cohen’s Kappa:

“Additionally, to ensure reliability, a randomized subset of 10 % of the full texts (n = 12) was independently screened by the author L.N. and the interrater reliability was calculated using Gwet’s AC1 [58] and the R package irrCAC [59].” (p. 10, l. 255 – 258)

“The interrater reliability of the study selection process was Gwet’s AC1 = .906, indicating high agreement [58,59].” (p. 13, l. 332 – 333)

*Pages 8 and 9, lines 233 through 255 (Methods, Data Extraction and Synthesis Method) - Please tell the reader if any attempt was made to retrieve missing data from investigator. If not, then this should also be

---

## [Decision Letter · Decision Letter 3]

27 Nov 2025

Exploring the association between housing insecurity and mental health among renters: A systematic review of quantitative primary and secondary studies

PONE-D-23-35202R3

Dear Dr. Talmatzky,

Please accept my apologies for the error that led to your paper being incorrectly returned to review following previous  acceptance.

As noted previously by a member of the editorial board, we’re pleased to inform you that your manuscript has been judged scientifically suitable for publication and will be formally accepted for publication once it meets all outstanding technical requirements.

Kind regards,

Marianne Clemence

Division Editor

PLOS ONE

Additional Editor Comments (optional):

Reviewers' comments:

Reviewer's Responses to Questions

**Comments to the Author**

1. If the authors have adequately addressed your comments raised in a previous round of review and you feel that this manuscript is now acceptable for publication, you may indicate that here to bypass the “Comments to the Author” section, enter your conflict of interest statement in the “Confidential to Editor” section, and submit your "Accept" recommendation.

Reviewer #3: All comments have been addressed

2. Is the manuscript technically sound, and do the data support the conclusions?

Reviewer #3: Yes

3. Has the statistical analysis been performed appropriately and rigorously? 

Reviewer #3: Yes

4. Have the authors made all data underlying the findings in their manuscript fully available?

Reviewer #3: Yes

5. Is the manuscript presented in an intelligible fashion and written in standard English?

Reviewer #3: Yes

6. Review Comments to the Author

Reviewer #3: GENERAL COMMENTS

Thank you for the opportunity to review this revised systematic review (manuscript # PONE-D-23-35202R3) now titled “Exploring the association between housing insecurity and mental health among renters: A systematic review of quantitative primary and secondary studies”. The authors have done a very nice job in responding to my previous comments. I only have two brief remaining comments, neither of which require any changes on the part of the authors.

1. Updated searches – If the handling editor has said that you do not have to update your searches, then okay.

2. Second (Instruments for assessing risk of bias/quality) – While I’m fine with your choice of instrument for the assessment of risk of bias/quality, the references you provided for the assessment of such that I believe you say were derived from the editor are outdated. Since that time, additional instruments have been developed, one of which is the MASTER scale, a scale that allows for the assessment of quality/risk of bias across multiple different study designs (See: 1. Stone JC, Glass K, Clark J, Ritskes-Hoitinga M, Munn Z, Tugwell P, et al. The MethodologicAl STandards for Epidemiological Research (MASTER) scale demonstrated a unified framework for bias assessment. J Clin Epidemiol. 2021;134:52-64; 2.Ahmed AI, Kaleem MZ, Elshoeibi AM, Elsayed AM, Mahmoud E, Khamis YA, et al. MASTER scale for methodological quality assessment: Reliability assessment and update. J Evid Based Med. 2024.)

7. PLOS authors have the option to publish the peer review history of their article (what does this mean? ). If published, this will include your full peer review and any attached files.

**Do you want your identity to be public for this peer review?** For information about this choice, including consent withdrawal, please see our Privacy Policy .

Reviewer #3: No

---

## [Editor Report · Acceptance letter]

PONE-D-23-35202R1

PLOS ONE

Dear Dr. Talmatzky,

I'm pleased to inform you that your manuscript has been deemed suitable for publication in PLOS ONE. Congratulations! Your manuscript is now being handed over to our production team.

Kind regards,

on behalf of

Dr. Andrew T. Carswell

Academic Editor

PLOS ONE